# Computational Metagenomics: State of the Art

**DOI:** 10.3390/ijms26189206

**Published:** 2025-09-20

**Authors:** Marco Antonio Pita-Galeana, Martin Ruhle, Lucía López-Vázquez, Guillermo de Anda-Jáuregui, Enrique Hernández-Lemus

**Affiliations:** 1Computational Genomics Division, National Institute of Genomic Medicine, Mexico City 14610, Mexico; marcopita99@gmail.com (M.A.P.-G.); martinruhle@gmail.com (M.R.); nutricionlucialv@gmail.com (L.L.-V.); 2Investigadores e Investigadoras por México, Secretaría de Ciencia, Humanidades, Tecnología e Innovación (SECIHTI), Mexico City 03940, Mexico

**Keywords:** computational metagenomics, microbiome, *16S* sequencing, bacterial genomics, computational tools, phylogenetic colocation, machine learning (ML)

## Abstract

Computational metagenomics has revolutionized our understanding of the human microbiome, enabling the characterization of microbial diversity, the prediction of functional capabilities, and the identification of associations with human health outcomes. This review provides a concise yet comprehensive overview of state-of-the-art computational approaches in metagenomics, alongside widely used methods and tools employed in amplicon-based metagenomics. It is intended as an introductory resource for new researchers, outlining key methodologies, challenges, and future directions in the field. We discuss recent advances in bioinformatics pipelines, machine learning (ML) models, and integrative frameworks that are transforming our understanding of the microbiome’s role in health and disease. By addressing current limitations and proposing innovative solutions, this review aims to outline a roadmap for future research and clinical translation in computational metagenomics.

## 1. Introduction

The human microbiome, a complex ecosystem of microorganisms, plays a fundamental role in host physiology and disease [1]. High-throughput sequencing technologies have provided unprecedented insights into its composition and function. However, the sheer volume and complexity of metagenomic data—characterized by high dimensionality, sparsity, and compositionality—present formidable analytical challenges that require robust computational solutions [2,3].

Microbial communities residing in and on the human body have a profound impact on host physiology, immunity, and metabolic processes [1]. The advent of next-generation sequencing (NGS) technologies, particularly whole-genome shotgun (WGS) sequencing and *16S rRNA* gene sequencing, has revolutionized our ability to profile the microbiome with high resolution [2]. These techniques have enabled researchers to move beyond traditional culture-based microbiology and investigate the microbiome in its native environment. However, the complexity of microbial ecosystems and the vast amounts of sequencing data generated necessitate sophisticated computational approaches for accurate characterization and interpretation [3].

Computational metagenomics involves the application of bioinformatics tools to process, analyze, and interpret metagenomic sequencing data. Key methodologies include reads clustering, genome assembly and binning, taxonomic classification, and functional annotation and comparative analyses across health and disease states [2]. Recent advances in machine learning (ML) and artificial intelligence (AI) have further enhanced the capacity to identify disease-specific microbial signatures, predict functional pathways, and model host-microbiome interactions [4]. These innovations hold immense potential for identifying microbial biomarkers, understanding disease etiology, and guiding therapeutic interventions [5].

Despite these advances, several challenges persist. The reliance on reference databases introduces biases in taxonomic and functional classification, and the high dimensionality of metagenomic data complicates statistical modeling and interpretation [3]. Additionally, integrating metagenomics with other omics data, such as transcriptomics, metabolomics, and proteomics, remains an open challenge in systems biology [4]. Overcoming these hurdles will be essential to fully harness the potential of computational metagenomics in translational research and clinical applications [2].

This article offers a comprehensive overview of state-of-the-art computational approaches in metagenomics, complemented by widely used methods and tools employed in amplicon-based metagenomics. Aimed at new researchers entering the field, it highlights key methodologies, challenges, and future directions, discussing recent advances in bioinformatics pipelines, ML models, and integrative frameworks that are reshaping our understanding of the microbiome’s role in health and disease [3]. By addressing current limitations and proposing innovative solutions, it outlines a roadmap for future research and clinical translation in computational metagenomics.

The content of this review is based on a comprehensive literature search conducted on PubMed, Google Scholar, Scopus, arXiv, and bioRxiv. We used Boolean combinations of English keywords mapped to the article’s sections to capture section-relevant literature—for example:

Overview of Metagenomics and Microbiome Analysis: “metagenomic sequencing”, “microbiome diversity and dynamics”, “microbial ecology and community structure”, “computational metagenomics”, “high-throughput environmental microbiomes”.

Computational Methods for Metagenomics and Microbiome Analysis: “shotgun metagenomics assembly & binning”, “*16S rRNA* sequencing pipelines”, “taxonomic profiling and read classification”, “genome-resolved metagenomics”, “gene prediction and functional annotation”.

Downstream Analysis: “functional profiling and pathway inference”, “differential abundance analysis”, “multi-omics integration (metagenome + metatranscriptome/metabolome)”, “compositional data analysis (log-ratio methods)”, “network and interaction analysis”.

Current use of metagenomics in Human Health: “microbiome and metabolic disease (obesity, diabetes)”, “microbiome-associated liver disease”, “maternal–fetal microbiome and pregnancy outcomes”, “gut–brain axis and neurological disorders”, “microbiome therapeutics (probiotics, prebiotics)”.

AI and ML in Metagenomics: “machine learning for microbiome classification”, “predictive modeling of treatment response”, “microbial feature selection and representation learning”, “functional trait prediction from metagenomes”, “explainable AI/interpretability in microbiome models”.

Data Sharing and Open Science: “FAIR microbiome data & metadata harmonization”, “sequence repositories and database curation”, “benchmarking of reference databases and tools”, “privacy, re-identification risk, and controlled access”, “minimal metadata standards”.

We applied the following inclusion criteria: peer-reviewed primary research and reviews; computational tools with active development or wide adoption; studies showing notable methodological advances or significant applications; and selective preprints addressing urgent gaps. Foundational papers were kept for context, but we prioritized literature from the last five years to provide a state-of-the-art perspective.

### 1.1. Overview of Metagenomics and Microbiome Analysis

Consider exploring the hidden world of microorganisms that profoundly influence our health and environment. Metagenomics is the study of the collective genetic material of microorganisms recovered directly from environmental samples, typically referring to DNA. However, it is important to note that RNA viruses can also be studied within metagenomic frameworks by converting their RNA into complementary DNA (cDNA) prior to sequencing. This approach broadens the scope of metagenomics, enabling the characterization of both DNA-based and RNA-based microbial communities and providing a more comprehensive view of their diversity and ecological roles. At its heart lies the microbiome—a dynamic community of bacteria, archaea, fungi, and viruses that thrive in specific environments, from the depths of the ocean to the human gut [1,2]. While the terms “microbiota” and “microbiome” are often used interchangeably, they hold subtle distinctions: microbiota refers to the living microorganisms themselves, such as those in our oral or gut ecosystems, whereas the microbiome encompasses their genetic blueprint and functional potential. In this review, we focus on the analysis of microbiota, particularly its role in human health and disease, and explore the state-of-the-art computational tools and methods that are transforming our understanding of these microbial ecosystems.

Unlike traditional culture-dependent methods, metagenomics enables the analysis of both culturable and unculturable microorganisms, providing a comprehensive view of microbial diversity and their functional potential [6]. Metagenomics relies on high-throughput sequencing technologies (e.g., next-generation sequencing) to generate vast amounts of sequence data from environmental samples.

### 1.2. Significance of Computational Approaches in Analyzing Microbial Communities

Metagenomic studies generate massive and complex datasets that require sophisticated computational tools for analysis and interpretation [3]. Computational approaches are thus crucial for tasks such as quality control (QC) of sequencing reads, assembly of DNA fragments into longer contigs, taxonomic classification of microbial sequences, and functional annotation of genes. These approaches enable researchers to identify microbial communities, predict their functional capabilities, and uncover their roles in various ecosystems [4]. Additionally, computational methods facilitate comparative analyses across different microbiomes, allowing for the identification of microbial signatures associated with specific environmental conditions or host phenotypes [2].

### 1.3. Recent Advancements in Computational Tools for Biological Research

The field of computational metagenomics has witnessed significant advancements in recent years, driven by the development of novel algorithms and bioinformatic tools [3]. There has been a surge in tools leveraging AI, particularly ML and deep learning, to handle the high dimensionality and complexity of metagenomic data [3,4]. These advanced tools provide more accurate taxonomic profiling, enhance functional predictions, and enable the identification of novel microbial biomarkers. Furthermore, the integration of multi-omics data (e.g., metagenomics, metatranscriptomics, metaproteomics) with computational models is facilitating a more holistic understanding of microbial communities and their interactions within complex ecosystems [7]. These advancements in computational metagenomics are playing a crucial role in accelerating biological research and unlocking the full potential of microbiome studies in areas such as human health, agriculture, food safety, and environmental science [5]. The field of computational metagenomics has witnessed significant advancements in recent years, driven by the development of novel algorithms and bioinformatic tools [3]. There has been a surge in tools leveraging AI, particularly ML and deep learning, to handle the high dimensionality and complexity of metagenomic data [3,4]. These advanced tools provide more accurate taxonomic profiling, enhance functional predictions, and enable the identification of novel microbial biomarkers. Furthermore, the integration of multi-omics data (e.g., metagenomics, metatranscriptomics, metaproteomics) with computational models is facilitating a more holistic understanding of microbial communities and their interactions within complex ecosystems [7]. These advancements in computational metagenomics are playing a crucial role in accelerating biological research and unlocking the full potential of microbiome studies in areas such as human health, agriculture, food safety, and environmental science [5].

## 2. Computational Methods for Metagenomics and Microbiome Analysis

In metagenomic research, as in many other scientific fields, experimental design plays a pivotal role, as it provides the framework to address specific biological questions or phenomena. The formulation of these questions not only guides the overall study design but also determines the sequencing methodology to be applied. The choice of sequencing technology, in turn, shapes critical downstream steps, from sample processing and library preparation to the selection of bioinformatic tools and computational strategies for data analysis. To establish a coherent narrative of a traditional metagenomic workflow, the following section offers a brief introduction to sequencing approaches, setting the stage for the subsequent discussion on bioinformatic tools. This decision cascade—beginning with research objectives and extending through sequencing choices—ultimately dictates the achievable taxonomic resolution, analytical depth, and resource requirements, as outlined in Figure 1.

### 2.1. Method Approach and Sequencing Technologies

#### 2.1.1. Technology Selection Framework

Short-read sequencing (SRS) using Illumina platforms remains the standard for cost-effective, high-accuracy sequencing, while long-read sequencing (LRS) technologies from Pacific Biosciences (PacBio) and Oxford Nanopore Technologies (ONT) offer superior assembly capabilities and structural variant detection [8]. The choice between these platforms creates distinct downstream analytical pathways, with each technology requiring platform-specific quality control, assembly algorithms, and taxonomic classification tools.

#### 2.1.2. Resolution Capabilities: Species Versus Strain-Level Analysis

The transition from technology selection to analytical capabilities reveals fundamental differences in taxonomic resolution potential. For species-level identification, shotgun metagenomics using either short- or long-read platforms generally outperforms *16S rRNA* sequencing, which struggles with species-level discrimination due to sequence conservation within certain bacterial taxa [8]. Recent benchmarking studies demonstrate that nanopore sequencing promises improved classification through longer reads [9], with LRS showing superior performance in detecting low-abundance species with high accuracy compared to SRS methods.

Strain-level resolution presents more complex challenges and requires careful consideration of both technology limitations and analytical approaches. While *16S rRNA* sequencing typically provides genus to species-level classification, strain-level differentiation demands either shotgun approaches with sufficient depth or targeted approaches focusing on variable genomic regions. Long read DNA sequencing allowed for the reconstruction of higher quality and even complete microbial genomes [10], making it particularly valuable for strain-level analyses that require detailed genomic context. However, strain resolution depends heavily on community complexity, sequencing depth, and the availability of high-quality reference genomes.

PacBio’s HiFi technology demonstrates very high accuracy, while ONT platforms generate ultra-long reads but with moderate base-calling accuracy in homopolymeric regions, making them suitable for assembly-focused applications where contiguity is prioritized over single-base precision [8]. The choice between these platforms directly impacts the sequencing depth requirements and bioinformatics workflows outlined in subsequent processing steps of Figure 1.

#### 2.1.3. Practical Experimental Design Recommendations

These technological capabilities translate directly into specific experimental design requirements that bridge the wet-lab and computational phases of metagenomic studies. Sequencing depth guidelines vary substantially depending on community complexity and host contamination levels. For shotgun metagenomics, typical ranges of 5–20 million reads per sample suffice for basic community profiling, while metagenome-assembled genomes (MAGs) recovery requires 50–200 million reads depending on target genome completeness and community diversity. Recent comparative studies suggest that hybrid strategies can provide enhanced resolution while balancing cost considerations [10]. LRS typically requires lower read counts (0.5–5 million reads) due to increased information content per read, though detecting rare taxa may require proportionally higher coverage.

DNA extraction considerations become critical when implementing LRS workflows, as these technologies are more sensitive to input DNA quality than traditional short-read approaches. High-molecular-weight DNA extraction is essential for optimal LRS performance, requiring specialized protocols that minimize shearing. Kim et al. [8] recommend avoiding extraction kits that fragment DNA below 50 kilobases, suggesting specific commercial kits including Circulomics Nanobind, QIAGEN Genomic-tip, or QIAGEN MagAttract HMW DNA systems. Sample handling protocols must minimize freeze–thaw cycles and avoid exposure to extreme pH conditions or intercalating dyes that can compromise long-read library preparation.

QC measures should include extraction blanks, PCR negatives for amplicon studies, and standardized mock communities for benchmarking taxonomic classification accuracy. For LRS applications, DNA integrity assessment using gel electrophoresis or automated fragment analysis systems ensures fragment sizes exceed 20 kilobases before costly library preparation steps.

Maximizing taxonomic resolution often benefits from hybrid approaches that combine the high accuracy of SRS with the assembly capabilities of LRS, leveraging the strengths of each technology while mitigating individual limitations [10,11]. For targeted strain-level analysis, supplementing *16S* data with alternative markers such as *gyrB* or *18S rRNA*, which exhibit higher variability than *16S* in specific taxonomic groups, can enhance resolution without the costs associated with shotgun approaches.

#### 2.1.4. Strategic Technology Selection

The optimal sequencing strategy depends on study scale, taxonomic targets, and analytical goals, with hybrid approaches offering particular promise for comprehensive microbiome characterization. Large-scale epidemiological studies typically benefit from cost-effective short-read approaches, while detailed mechanistic studies requiring precise genome reconstruction should prioritize long-read technologies or hybrid strategies that combine both approaches strategically. Table 1 provides a comparative framework for selecting appropriate sequencing technologies based on study-specific requirements and resource constraints.

### 2.2. Bioinformatics Pipelines

Different bioinformatics pipelines are tailored for various microbiome sequencing approaches, including marker gene (*16S*, *18S*, ITS) sequencing and WGS metagenomics. These workflows are summarized in Figure 2. This section outlines these computational workflows, starting from raw sequence processing to data-ready outputs for downstream analysis.

#### 2.2.1. Computational Paths: From Raw Data to FASTQ

The first step in most bioinformatics pipelines is converting raw data into FASTQ format, where reads are retained with quality information for analysis. Depending on the sequencing platform, initial processing may include base calling, quality score calibration, and demultiplexing:

Nanopore: Raw files (FAST5) processed into FASTQ format through EPI2ME, offering tools for real-time analysis [12].

PacBio: Generates long reads suitable for de novo assembly, with processing performed using the SMRT Portal [13].

Illumina: Converts BCL files to FASTQ using bcl2fastq, typically generating short, high-quality reads (Illumina) [14].

#### 2.2.2. Preprocessing and Quality Control

QC and preprocessing steps help reduce errors and bias, ensuring reliable downstream analysis. For each sequencing method, specific tools and workflows are recommended to filter, trim, and refine the sequence data:

Quality Filtering: FastQC assesses read quality distributions [15], while tools like Trimmomatic [16] and Fastp [17] filter low-quality bases.

Adapter Trimming: Cutadapt [18] and Porechop [19] identify and trim adapter sequences.

Comprehensive Platforms: Fastp combines several preprocessing tasks, offering a high-speed and effective solution for general sequence QC and adapter trimming.

Combining tools often yields the best QC results; for example, FastQC for read quality assessment, followed by cutadapt for adapter trimming, and then Trimmomatic for low-quality base filtering.

#### 2.2.3. Contig/Genome Assembly Approaches

Once high-quality reads are obtained, genome assembly is performed to reconstruct microbial genomes or contigs. Assembly strategies depend on both the type of sequencing data and the study objectives. Two main approaches are used:De novo Assembly:

Assemblers reconstruct genomes without relying on reference sequences. Tools such as MEGAHIT [20] and metaSPAdes [21]—which use de Bruijn graph algorithms—are widely employed for short-read metagenomic data. IDBA-UD [22] offers an iterative de Bruijn graph approach that is particularly effective for handling the uneven sequencing depths commonly encountered in complex metagenomic communities. For LRS data, Flye [23] employs an overlap-layout-consensus strategy that can produce highly contiguous assemblies, though it requires substantial computational resources. De novo methods excel in exploring diverse and unknown environments but often yield fragmented assemblies due to uneven coverage and high strain diversity.
Reference-guided Assembly:

This approach aligns reads to existing reference genomes using tools like BWA [24] and Bowtie2 [25]. It provides higher resolution for taxa with well-characterized genomes; however, it is limited when novel organisms are present.

In many cases, hybrid strategies that combine de novo and reference-guided techniques are adopted to maximize assembly quality. To facilitate comparisons among assembly methods, Table 2 outlines the five most important tools, selected based on their citation frequency in the recent literature, algorithmic novelty, and demonstrated performance improvements over existing methods. While other innovative tools may exist, they have not yet been sufficiently benchmarked to be considered.

#### 2.2.4. Binning of Metagenomic Contigs and Genome Recovery

Because de novo assemblies are often fragmented, binning methods group contigs originating from the same or closely related organisms. These methods fall into three categories [26]:Composition-based Methods:

Rely on nucleotide features such as k-mer frequencies and GC content. Early tools (e.g., TETRA, CompostBin) assume that sequences from the same taxonomic group have similar oligonucleotide signatures.
Abundance-based Methods:

Use read coverage information, since contigs from the same genome should exhibit similar abundance across samples. Tools like MetaBAT 2 [27] and MaxBin 2 [28] are prime examples.
Hybrid Methods:

Combine both composition and abundance features for more robust binning. Recent approaches leverage advanced ML techniques, such as variational autoencoders in VAMB [29] and contrastive multi-view representation learning in emerging tools like COMEBin [30]. Additionally, semi-supervised deep learning approaches like SemiBin2 [31] demonstrate how leveraging both labeled reference genomes and unlabeled metagenomic data can significantly improve binning performance across diverse environmental samples.

For a clearer comparison of binning methods, Table 3 outlines the five most important tools using the same criteria as for Table 2. This selection highlights emerging methods that underscore the dynamic nature of the field, while acknowledging that other innovative tools may exist but have not yet been sufficiently benchmarked to be considered.

#### 2.2.5. Evaluation of MAG Quality

The final step in the reconstruction pipeline is to evaluate the quality of the MAGs. Quality assessment is critical because it determines the reliability of downstream analyses [32]. The main metrics include:Completeness and Contamination:

Tools such as CheckM [33] use lineage-specific marker genes to estimate the proportion of a genome that is recovered (completeness) and the presence of contaminant sequences (contamination). BUSCO [34] extends this evaluation by assessing the presence of conserved single-copy orthologs across various taxonomic groups. The Minimum Information about a Metagenome-Assembled Genome (MIMAG) standards are the community-accepted benchmark, classifying MAGs as either medium-quality (≥50% completeness, <10% contamination) or high-quality (≥90% completeness, <5% contamination, and presence of essential *rRNA* and *tRNA* genes) [35].
De-replication and Cataloging:

Tools like dRep and Mash facilitate the de-replication of MAGs by comparing genomes based on average nucleotide identity (ANI) to ensure a non-redundant, high-quality catalog. Such steps are essential when pooling results from multiple samples or assemblies.
Additional Metrics:

Other important indicators include the number of contigs per MAG, strain heterogeneity, and the presence of essential genes such as *16S*, *23S*, and *5S rRNA* as well as a sufficient number of tRNAs. These metrics are part of the MIMAG standards mentioned above [27].

It is crucial for researchers to recognize that a significant fraction of bins generated from complex samples, such as the gut or soil microbiomes, may fail to meet these stringent quality thresholds. Factors like high strain-level diversity, the presence of repetitive genomic regions, and low sequencing coverage for many community members can result in fragmented assemblies, which in turn leads to the generation of MAGs with low completeness [32]. This reality presents a major challenge for downstream biological interpretation.

Using low-completeness MAGs for downstream inference carries significant risks. A MAG that is only 50–70% complete is missing a substantial portion of its gene content, which makes any functional annotation speculative and potentially misleading. For example, crucial metabolic pathways may appear absent simply because the genes encoding them were not recovered in the assembly. Furthermore, the absence of key phylogenetic marker genes can lead to inaccurate taxonomic placement [36]. Therefore, for reliable functional profiling and comparative genomics, it is strongly recommended that researchers filter their MAG collection based on established quality standards (e.g., MIMAG medium- or high-quality) to ensure that biological conclusions are drawn from robust and reasonably complete genomic data.

Together, these evaluation steps help researchers select the most reliable MAGs for downstream analyses such as functional annotation, phylogenetic reconstruction, and biomarker discovery.

#### 2.2.6. Reads Clustering for Taxonomic Classification

Accurate taxonomic classification of metagenomic sequences is fundamental for understanding microbial community structure. Two main approaches are used:

Marker Gene-Based Clustering:

For targeted taxonomic profiling, marker genes like *16S rRNA* are amplified and sequenced. Tools such as QIIME2 [37] and DADA2 [38] process these sequences into operational taxonomic units (OTUs) or amplicon sequence variants (ASVs), providing high-resolution taxonomic assignments for known groups.

WGS Classification:

In shotgun metagenomics, taxonomic labels are assigned directly to raw reads using k-mer–based classifiers. Tools such as Kraken 2 [39] and Centrifuge [40] rapidly compare read k-mer signatures against extensive reference databases, offering broad coverage of the microbial diversity—including rare and novel taxa.

Integration with Binning and Downstream Analysis:

Taxonomic classification is further refined by integrating binning results. Once contigs are binned and MAGs are reconstructed, tools like GTDB-Tk assign taxonomic labels to the bins by placing them within a standardized taxonomic framework. This integrative approach not only validates the binning outcomes but also enhances the resolution of microbial community analyses.

High-resolution DNA fragments clustered into MAGs or ASVs representing marker genes form the essential foundation for converting molecular fragments into a close approximation of biological reality, such as taxonomic and functional profiles. The main purpose of these profiles is to explore ecological and biological hypotheses through the integration of microbial identity and function. In the following section, we provide an approximation to the most widely used tools for analyzing this aspects of a microbiome.

#### 2.2.7. Taxonomic Profiles

An initial step in interpreting the composition of a microbiome is to determine which organisms it contains. The primary concept behind a taxonomic profile is to group the most similar sequences from a sample to a particular taxon, most often by matching them against reference databases. The choice of profiling strategy or tool is generally dictated by the sequencing methodology employed.

##### Amplicon Metagenomics

Amplicon based metagenomics offer a rapid and cost-effective strategy for human microbiome analysis, with low sequencing costs (usually around USD 20–50 per sample) and small file sizes. Because only a small portion of the genome is being sequenced, taxonomic profiling can be performed with relatively low computational demands using local alignment algorithms such as UPARSE [41] and BLAST [42], or k-mer based approaches such as DADA2 [37], commonly implemented in pipelines like QIIME 2 [37]. However, these methods generally achieve resolution only to the genus level because marker genes—such as the *16S rRNA* gene—cannot fully capture evolutionary differences among all bacteria. Additionally, copy number variation in marker genes, and sequence heterogeneity between operons within the same species, can introduce false positives during taxonomic assignment [43,44].

##### Whole Genome Metagenomics

In recent years, whole genome metagenomics has gained prominence due to improvements in accuracy of both short and long sequencing technologies as well as the rapid expansion and enrichment of reference databases. This has promoted the production and enhancement of numerous tools that allow taxonomic classification and profiling. Short reads oriented tools such as Bracken [45], Kraken 2 [39], Centrifuge [40] and CLARK/CLARK-S [46,47] perform k-mer based matching against reference databases. This strategy enables high-throughput processing of large datasets (<1 h) if high memory computational resources are provided (>100 GB of memory).

Across tools, Kraken 2 has proven to offer an effective balance between speed and accuracy. On the CAMI2 gastrointestinal dataset [48] it achieved a precision of 0.94 coupled with fewer than 1% false positive and an impressive F1-score of 0.97, highlighting its robustness for species level assignments. In parallel, Centrifuge reached a very similar precision of 0.93, however it exhibited a rate of false positives around the 3% rate, while still obtaining a 0.97 F1-score [49]. From a practical standpoint, Kraken 2 may be favored in scenarios where minimizing false positives is critical, such as clinical diagnostics or studies requiring stringent taxonomic confidence. In contrast, Centrifuge offers advantages in settings where maximizing recall and broad detection are priorities, for instance in exploratory surveys of highly diverse or poorly characterized communities, even at the cost of a modest increase in false positives.

Although these tools can achieve species level resolution, their k-mer based approach often leads to substantial generation of false positives when the sample contains closely related taxa or organisms present at very low abundance (~0.05–1%). To mitigate this, researchers often apply abundance thresholds, focusing on retaining only taxa with relative abundances >1% depending on study objectives.

When computational resources are limited, marker-based approaches such as MetaPhlAn [50] mOTUs [51], are recommended. These tools rely on databases comprising a large number of gene families rather than complete reference genomes; however, the absence of full genome context can result in a high proportion of false positives. For this reason, the use of the most recent versions of these tools is advised. mOTUs 3 [52] incorporates 5165 metagenomes from human associated microenvironments and substantially improves species level precision and recall, obtaining a F1-score ranging from 0.78 to 0.85 across several CAMI2 human microbiome datasets [48]. MetaPhlAn 4 [53] expands its database to 19.5 metagenomes from diverse body sites, which has allowed it to obtain F1-scores of 0.80 to 0.85 when using the OPAL framework [54] and a nearly perfect 0.95 score when employing genomes from the SGB organization as reference when this method is applied in several CAMI2 human microbiome datasets [48]. Furthermore, the latest release (v4.2.2) of MetaPhlAn 4 adds support for long-read sequence profiling through Minimap2-based alignment.

#### 2.2.8. Gene Prediction and Annotation

Following taxonomic classification, genes are predicted and functionally annotated to understand microbial activity within communities. Annotation involves assigning function to gene sequences based on reference databases:

##### Gene Prediction

Tools like Prodigal [55] are widely used to identify genes in assembled contigs, being highly optimized for bacterial and viral genomes, and thus remain the preferred choice in prokaryote-dominated microbiomes. Prodigal has consistently shown high accuracy in benchmark datasets, achieving >90% sensitivity and specificity in bacterial genomes and CAMI community challenges [48]. However, when eukaryotic microorganisms such as fungi constitute a significant component of the community, specialized tools are more suitable. For example, MetaEuk [56], benchmarked on large-scale marine microbiomes, demonstrated high sensitivity (~92%) and precision (>85%) for gene recovery from fragmented assemblies by leveraging homology-based searches. Similarly, EukMetaSanity [57], tested on simulated fungal metagenomes and environmental datasets, integrates multiple predictors and achieved robust gene annotation with F1-scores of ~0.85–0.90. Therefore, the choice of software should be guided by both the taxonomic composition of the dataset and its complexity: Prodigal for prokaryotes and viruses, and MetaEuk/EukMetaSanity when eukaryotic members such as fungi or protists are of interest.

##### Functional Annotation

Functional annotation, as described by Wang [58], is a computational process that assigns putative functions to predicted genes discovered during gene discovery. Shotgun metagenomics enables the reconstruction of contigs and bins, facilitating detailed functional analysis of microbial communities. In contrast, marker gene approaches primarily provide taxonomic information and can only infer potential functions due to limited genomic data. Therefore, while marker gene methods offer insights into community composition, shotgun metagenomics is more effective for accurately characterizing the functional potential of microbiomes.

Marker gene approaches primarily offer taxonomic insights but are limited in their ability to determine functional capabilities due to the scarcity of genomic data. To address this limitation, various computational tools have been developed to leverage marker gene information aiming to bridge the gap between taxonomic profiling and functional characterization by predicting metabolic pathways and biological processes based on marker gene distributions. In this regard, tools like PICRUSt2 [59], Tax4Fun2 [60], BugBase [61] and PanFP [62] have been developed. When evaluating their accuracy, however, clear differences emerge. In simulated datasets, PICRUSt2 demonstrated one of the lowest false positive rates (FPR), averaging 2.94%, while MicFunPred also performed conservatively with an average FPR of 6.86%. In contrast, Tax4Fun2 exhibited the highest level of false predictions, with an average FPR of 35.14%, highlighting its tendency to overestimate functional content [63]. These results emphasize that while marker gene-based approaches can provide functional predictions, their reliability varies considerably across tools and should be interpreted with caution.

It is important to remark that these tools are inherently limited by their reliance on *16S* gene profiles and reference genomes. These methods extrapolate potential gene content based on phylogenetic similarity, as is the case of PICRUSt2 and Tax4Fun2 or pangenome reconstructions as PanFP, assuming that closely related taxa necessarily share similar functional capabilities. However, this approach overlooks intra-species variability, horizontal gene transfer events and strain specific adaptations. As a result, the inferred profiles represent approximations rather than measures of genetic content.

In contrast, whole genome sequences are assembled, to be further functionally annotated by comparing coding sequences with databases containing information on genes, proteins, and metabolic pathways [64]. This approach thus enables a comprehensive evaluation of the functional characteristics of microbial communities as they analyze the actual content of genes, and pathways present in a sample [65,66,67].

In this regard, tools like Prokka [68] and InterProScan [69] have been tested to annotate test genomes, which are capable of annotating between 75 and 90% of the individual genomes in less than 3 min. However, the reason for this performance is due to the limited database they use as reference, which can be an inconvenience when the samples analyzed contain multiple metagenomes. In contrast, tools like MicrobeAnnotator [70], which has higher computational demands, is able to annotate ~92% of individual genomes and can be used in complex samples as it combines multiple databases, offering an integral functional annotation of metagenomes.

Some tools like BlastKOALA and GhostKoala [71] are built as web services which offer different insights into genomic and metagenomic data. BlastKOALA delivers accurate annotations at the genome level, but its computational cost is substantial, requiring up to 80 min per genome. In contrast, GhostKOALA offers a much faster alternative, completing the annotation of full genome in less than 7 min. This gain in speed, however, comes at the expense of sensitivity, as GhostKOALA may miss novel functions or provide less comprehensive coverage compared to its counterpart.

Finally, alignment-based engines such as DIAMOND [72] and MMseqs2 [73] serve as the backbone for many pipelines. While DIAMOND delivers BLAST-like sensitivity at higher speeds, it loses sensitivity when dealing with highly divergent sequences or poorly conserved genes. MMseqs2 is even faster in large-scale comparisons, offering scalable annotation of millions of sequences with reduced memory footprints, yet this gain in efficiency can come at the cost of sensitivity, requiring careful parameter tuning and greater technical expertise to achieve optimal results.

Looking ahead, future work should prioritize systematic benchmarking of functional annotation tools such as InterProScan, Prokka, and Diamond. While taxonomic profiling has benefited from community-driven efforts like CAMI, functional annotation still lacks standardized large-scale evaluations with well-defined gold standards. Establishing such benchmarks would not only clarify the trade-offs in precision, recall, and F1-score across different approaches, but also provide researchers with evidence-based guidelines for tool selection depending on study goals. Ultimately, community-wide performance assessments could drive methodological improvements, foster the development of integrative pipelines, and enhance the reliability of functional insights derived from metagenomic data.

Newly developed but not yet widely adopted tools may offer innovative approaches, improved accuracy, or enhanced computational efficiency, addressing limitations of current methods and opening opportunities for more comprehensive functional and taxonomic analyses. For instance, DeepFRI [74] leverages deep learning strategies to perform functional annotations, achieving ~70% concordance with eggNOG, a widely recognized annotation framework. Alternatively, tools such as FlaGs [75] and FunGeCo [76] explore complementary strategies, including gene neighborhood conservation and functional genomic context, which may provide additional layers of biological insight beyond traditional annotation pipelines.

Although functional analysis serves as an inference of the possible metabolic capabilities of the studied microbiome, they can help in the hypothesis generation in newly studied populations. However, when describing/generating this hypothesis a few pointed assumptions and limitations must be considered.

Firstly, short read metagenomic approaches assume that these short sequences contain sufficient information to be mapped to its origin gene or an equivalent ortholog and, therefore, a correctly associated function. Secondly, these approaches assume that sequences obtained from intergenic regions or from orthology groups with unknown function will not be erroneously mapped [77]. Given that up to date most metagenomic studies approaches still rely on these approaches for large scale studies due to their cost-effectiveness and high accuracy [78,79], these assumptions must be carefully considered, as they propagate to downstream analysis [77].

That said, modern metagenomic workflows such as DRAM and HUMAnN3 as part of bioBakery 3 used to process short read sequences, prioritize assembly and the reconstruction of MAGs before carrying out functional annotation and profiling. This strategy markedly reduces annotation ambiguity, since contigs and MAGs provide a broader genomic context than individual reads, thereby improving functional assignment [36]. In addition, the use of long reads has enabled the capture of full length gene sequences and even entire operons owing to the generation of longer contigs during assembly. This has significantly enhanced the recovery of near complete and high quality MAGs, which are more informative for downstream functional analysis. Nevertheless, the elevated cost and infrastructure demands of LRS can limit its widespread adoption, particularly in large scale or clinical studies. To address this limitation, hybrid approaches such as hybridSPADES [80], OPERA-MS [81] and MetaPlatanus [82] have been increasingly recommended, as they combine the cost effectiveness and depth of short reads with the assembly advantages of long reads, facilitating the recovery of longer contigs and more complete MAGs but also improve the overall mapping rates to bacterial genomes, providing a more reliable foundation for functional characterization [10,79].

It is important to highlight that, even high quality MAGs are subject to errors that can arise from DNA extraction all the way to the assembly processes, leading to the loss of sets of genes and the overall representation of them in a particular microbiome [78]. Furthermore, although functional databases have significantly expanded in recent years, a substantial proportion of genes in microbiomes remain annotated as “hypothetical” [83]. This implies that functional inference is still largely dependent on homology to orthologous genes from reference organisms, many of which are biased towards well-studied eukaryotes or a limited set of prokaryotic models. Consequently, while functional profiling is a powerful tool for exploring metabolic potential, it is essential to acknowledge its inferential nature, the risks of over-interpretation, and the propagation of annotation errors in complex or under-characterized microbiomes.

Moving forward, the most important direction for refining functional annotation lies in the integration of high quality MAGs with long read and hybrid assemblies, coupled with the continuous expansion and curation of functional databases to reduce the proportion of genes classified as hypothetical. In parallel, the incorporation of complementary omics approaches such as metatranscriptomics, metaproteomics, and metabolomics will be critical to validate gene functions beyond sequence homology, thus bridging the gap between inferred metabolic potential and experimentally supported functional activity.

Given the diversity of strategies available for microbial community analysis, both taxonomic profiling and functional annotation tools have been extensively benchmarked to evaluate their accuracy, scalability, and robustness. To facilitate comparisons among such methods, Table 4 summarizes representative tools that have been widely tested in previous studies. While additional promising tools are continuously being developed, only those with sufficient benchmarking evidence have been included here.

#### 2.2.9. Metagenomic Pipelines

The standardization of bioinformatic workflows constitutes a fundamental strategy to maximize the use of existing tools, minimize human error, and ensure reproducible and scalable analyses. Initiatives such as MetaWRAP [88], MOCAT2 [89] or IMP [90] have integrated modules for read processing, bin generation and refinement, and data visualization, thereby expanding the scope of functional and multimodal analyses, although their adoption may be limited by complex configurations or technical constraints.

In response, the incorporation of workflow management systems such as Snakemake [91] and Nextflow [92]) has enabled the chaining of taxonomic profiling and functional annotation tools, thereby enhancing reproducibility, portability, and scalability, as well as increasing the overall data processing capacity.

A recent example is MaLiAmPi [93], a pipeline developed in Nextflow that harmonizes heterogeneous *16S* amplicon sequences from different studies through the generation of phylotypes, offering species and even subspecies level resolution. This approach has demonstrated superior predictive power compared to traditional methods such as QIIME2, albeit at a higher computational cost in both time and memory.

For whole-genome metagenomics, pipelines such as ATLAS [94] provide an accessible workflow managed through Snakemake, capable of generating MAGs with consistent levels of completeness (>60%) and low contamination (<10%). Its standardized design offers clear outputs that integrate species-level resolution together with abundance tables and functional profiles in a user-friendly format. However, this simplicity and robustness may limit its flexibility when addressing more exhaustive or customized analytical needs.

In contrast, SnakeMAGs [95] also implemented in Snakemake, produces MAGs of comparable quality but differs by prioritizing the quantity of MAGs recovered over workflow standardization. In a dataset of 10 metagenomes, SnakeMAGs recovered approximately 76% more MAGs than ATLAS, underscoring its capacity to maximize genomic diversity; nevertheless, this increase comes at the cost of longer execution times, highlighting a trade-off between comprehensiveness and computational efficiency.

On the other hand, tools such as SnakeWRAP [96], likewise implemented in Snakemake, focus on scalability and the management of large volumes of data, adopting rigorous quality practices that ensure reproducibility and portability. Although it does not integrate functional annotation within its workflow, its strength lies in the ability to reliably automate the processing of multiple datasets in parallel, positioning it as a preferred option in high-throughput scenarios, albeit with the limitation of providing less comprehensive results compared to more integrative pipelines.

The complexity inherent to the wide variety of taxonomic and functional profiling tools inevitably adds an additional layer of consideration during the design of an investigation, since the selection of one approach over another directly impacts the computational infrastructure needed and inevitably influences the resolution and accuracy of the resulting data. Equally important, however, is the choice of the reference database, which represents a critical determinant of the quality, scope, and interpretability of the taxonomic and functional information extracted from sequencing data. In the following section, we examine the main databases commonly employed for this purpose, highlighting their specific features and advantages.

### 2.3. Choosing an Adequate Database

Databases represent the cornerstone upon which profiling tools operate, as they provide the curated taxonomic and functional references required for reliable analyses. Their selection is therefore decisive, since well-suited databases minimize false positives and enhance the accuracy of both taxonomic classification and functional annotation [86,97,98,99,100,101,102].

As Martins [103] has stated, metagenomic databases can be organized and classified following different criteria, to this end, and only contemplating the human biomedicine field, we present a group of common databases used to answer biological questions of interest as follows:

#### 2.3.1. Sequence Repositories

Public repositories play a crucial role in metagenomic research by providing access to extensive datasets including raw reads, complete genomes, and metagenomes. These repositories are enriched with associated metadata from diverse case studies and cohorts, representing specific health conditions. By offering standardized and openly accessible data, public repositories enhance the reproducibility of analysis and facilitate the integration of additional datasets, ultimately enabling more robust and comprehensive investigations in metagenomics.

In addition to their significance in metagenomic studies, publicly accessible nucleotide sequence repositories are essential for data preservation and global information sharing. Among the most widely used repositories are the European Nucleotide Archive (ENA) [104], the repositories of the National Center for Biotechnology Information (NCBI), namely Genbank [105] and the Sequence Read Archive (SRA) [106] as well as the DNA Data Bank of Japan (DDBJ) [107] as part of the Institute of Genetics in Mishima, Japan. These repositories collectively form the International Nucleotide Sequence Database Collaboration (INSDC) [108] which ensures that all participating databases remain synchronized containing identical and up to date set of sequence records while adhering to shared standards for data format and annotation.

The ENA, maintained by the European Molecular Biology Laboratory’s European Bioinformatics Institute (EMBL-EBI) provides an open platform for managing and disseminating nucleotide sequencing data, offering a globally comprehensive collection of sequences with robust tools for data submission and retrieval. At NCBI, Genbank serves as the central repository of annotated DNA sequences while the SRA is the world’s largest archive of raw high-throughput sequencing data. Similarly, the DDBJ acts as a primary sequence repository while also managing specialized services within the INSDC including the Bioproject platform which provides sequencing project metadata and Biosample, which ensures comprehensive sample information [109].

#### 2.3.2. General Databases for Taxonomic Analysis

General databases for taxonomic analysis utilize genetic information to classify organisms based on their taxonomic identity. They provide genomic data across various domains to address a wide range of questions related to the abundance and diversity of microorganisms. Databases such as SILVA, GreenGenes, NCBI Taxonomy, and UNITE contain sequence data from marker genes like *16S*, *18S* and ITS.

SILVA [110] is a comprehensive online resource that provides the user high-quality and updated datasets of aligned small (*16S*/*18S*) and large subunit (*23S*/*28S*) ribosomal RNA sequences integrated from several external resources and it includes sequences from the domains Bacteria, Archaea and Eukarya. On the other hand, Greengenes [111] is a comprehensive, high-quality database which contains *16S rRNA* reference sequences with taxonomy based on a de novo phylogeny.

UNITE [112] is a specialized database focused on the internal transcribed spacer (ITS) region of *rRNA* sequences, primarily targeting fungi. It provides reference sequences for the identification and classification of fungi and is an essential tool for studies on fungal diversity and ecology.

#### 2.3.3. General Databases for Functional Analysis

Functional analysis involves assigning putative functions to previously predicted genes using various strategies to elucidate metabolic capabilities of microbial communities. This process relies on various databases that group gene orthologs and provide detailed representations of metabolic pathways. The most prominent databases utilized in this process are mainly composed by eggNOG [113], which offers hierarchical, non-supervised orthologous groups constructed from numerous genomes across different domains of life. The Cluster of Orthologous Genes databases (COG) [114] classifies proteins from complete genomes into orthologous groups to reflect evolutionary relationships. Each COG includes proteins that are thought to be derived from a common ancestor, allowing for the prediction of gene functions based on orthology.

KEGG [115] is a comprehensive resource that integrates genomic, chemical and systemic functional information, providing detailed representation of metabolic pathways, molecular interactions and cellular processes, enabling the mapping of genes and proteins to specific pathways. UniProt [116] serves as a central hub for protein data, supporting functional annotation and the study of protein roles in various biological processes. MetaCyc [117] is a curated database of experimentally elucidated metabolic pathways from all domains of life, useful for metabolic reconstruction and analysis. Gene Ontology provides a framework for the representation of gene function, detailing biological processes, cellular components, and molecular functions.

#### 2.3.4. Particular Human Microbiome

The environment significantly influences the composition and function of microbial communities, a principle evident in various human microhabitats. Utilizing ecosystem-specific datasets as references has advanced our understanding of the unique microbiome structures within these niches. The mainly used databases used to this end are the gutMEGA [118] database is a comprehensive resource that compiles metagenomic data specific to the human gastrointestinal tract. It provides a collection of assembled genomes and gene catalogs derived from gut microbiome samples, facilitating studies on microbial diversity, function, and their associations with human health. The Unified Human Gastrointestinal Genome (UHGG) collection [119] is an extensive collection of reference genomes from the human gut microbiome. The initial release comprised 204,938 non-redundant genomes representing 4644 gut prokaryotic species. These genomes encode over 170 million protein sequences, organized into the Unified Human Gastrointestinal Protein (UHGP) catalog.

The Human Oral Microbiome Database [120] provides comprehensive information on the bacterial species present in the human oral cavity. It includes genomic sequences, taxonomic classifications, and metadata related to oral bacteria. This database supports research into oral microbiome composition, its role in oral health and disease, and its interactions with other human microbiomes. The skin microbiome database (SKIOME) [121] focuses on the diverse microbial communities inhabiting human skin. It offers genomic data, taxonomic information, and functional annotations of skin-associated microorganisms. SKIOME aids in understanding the skin microbiome’s role in health, disease, and its response to various environmental factors. The Vaginal Microbiome Consortium [122] is an initiative that compiles data on the microbial communities of the human vaginal environment. It provides genomic sequences, taxonomic classifications, and functional annotations pertinent to vaginal microbiota.

#### 2.3.5. Specialized Questions

When researchers address specialized questions in metagenomics, the utilization of meticulously curated and specific databases becomes essential. These specialized resources enable precise identification and functional characterization of genes related to particular biochemical processes or resistance mechanisms, some of the most commonly used databases employed for such specialized analyses are the Novel Metagenome Protein Families Database (NMPFamsDB) [123] which focuses on the discovery of novel protein families identified from de novo assembled microbial genomes with no similarity to known protein motifs or domains which are later clustered in families that have environmental and taxonomic metadata. The Carbohydrate-Active enZYmes (CAZy) database [124] is a specialized resource which classifies enzymes involved in the synthesis, metabolism and recognition of complex carbohydrates, classifying them into families and provides information about their function and associated biological processes.

Some researchers focus on the identification and understanding of antibiotic resistance genes to address the worldwide antimicrobial resistance problem. To address these questions, several specialized databases have been developed like the Comprehensive Antibiotic Resistance Database (CARD) [125] which provides highly and continuously updated hand curated reference DNA and protein sequences related to resistance variants. The MEGARes database [126] is a compendium of several reference sequences for genetic determinants of resistance to drugs, metals and biocides which contains approximately 8000 manually curated resistance genes that are adapted to high-throughput classification due to its acyclic hierarchical annotation structure. Resfams [127] is a curated database of protein families with confirmed antimicrobial resistance functions that, due to their organization based on ontology, aids in the accurate annotation of antimicrobial resistance genes.

## 3. Downstream Analyses in Metagenomics

As a final output, metagenomic tools and pipelines generate count tables containing taxonomic and functional information, which serve as the essential input for downstream analyses aimed at interpreting their biological and ecological significance (Figure 3), particularly in highlighting subtle differences between health and disease states [128]. This section outlines the key approaches to downstream analysis, encompassing descriptive and inferential statistical analysis, multi-omics data integration, and network-based interaction analyses.

### 3.1. Descriptive Analysis

The exploratory phase involves analyzing diversity metrics and visualizing them using ordination plots—a term in ecology that encompasses various multivariate techniques for representing species abundance in a low-dimensional space.

Diversity is described in terms of within-sample diversity (alpha diversity) and between-sample diversity (beta diversity); The alpha diversity is mainly explained using estimators like the Shannon index [129] which explains diversity in terms of the number of species and their proportion in a sample, the Chao1 index [130] which describes the number of unobserved/low abundance species in the samples or the Faith’s phylogenetic diversity index [131]. Unlike alpha diversity, beta diversity measures the compositional differences between samples. The most commonly used metrics for this purpose are based on distance matrices, such as the Bray–Curtis dissimilarity index [132] and the UniFrac distances [133].

Effective visualization of alpha and beta diversity is essential for interpreting microbial community structure and composition. Bar plots provide an intuitive representation of relative taxa abundances (Figure 3A) and the distribution of diversity indexes (Figure 3B), while heatmaps reveal complex patterns of co-occurrence across samples [134,135,136]. A common approach for the visualization of beta diversity is the use of ordination methods, multivariate techniques that allow species abundance to be visualized in a low-dimensional space, such as principal component analysis (PCA) and multidimensional scaling (MDS) (Figure 3C) [137] are available through R packages such as vegan [138], ade4 [139] and phyloseq [140].

### 3.2. Statistical Analysis

Metagenomic data exhibits several particularities that complicate statistical modeling. First, these datasets are inherently high dimensional, as they encompass a large set of microbial taxons or functional features across relatively few samples. This is compounded by their sparse structure, with abundance matrices inflated by zeros, due to the heterogeneous distribution of these features across hosts and the limited sequencing depth. Moreover, when features are detected, their counts usually exhibit a large fluctuation across samples, leading to over-dispersion, in which the variance greatly exceeds the mean. In addition, as sequencing yields relative rather than absolute abundances, microbiome data are fundamentally compositional, constraining the data to a constant sum space and inducing spurious correlations.

#### 3.2.1. Differential Abundance

The simultaneous presence of high dimensionality, sparsity, over dispersion and compositionality poses significant challenges for downstream statistical analysis, requiring specialized frameworks to ensure robust and biologically meaningful description and inference. Therefore, appropriate data normalization constitutes a crucial step to account for the characteristics of microbiome data and improve the comparability of samples across data.

There are several strategies employed in normalization methods, some are based on data scaling such as Total Sum Scaling (TSS), Cumulative Sum Scaling (CSS), Trimmed Mean of M-Values (TMM) and Relative Log Expression (RLE) which adjust samples count data by a normalization factor to correct the different sequencing depth or global composition, and others like Centered-Log Ratio (CLR) and Variation Stabilizing Transformation (VST) transform relative abundances to a logarithmic space, eliminating the constant sum restriction.

After normalization, differential abundance analysis aims to test whether microbial features differ significantly and the magnitude of change between clinical in their abundance across phenotypes or experimental groups (Figure 3D). To this end, several R packages like DESeq2 [141], LEfSe [142], edgeR [143], ALDEx2 [144] and ANCOM-BC2 [145], are commonly used.

For instance, DESeq2 and edgeR, originally developed for differential gene expression in RNA-Seq, are now frequently applied in microbiome studies. These tools leverage RLE and TMM normalization, respectively, and model counts using the Negative Binomial distribution to accommodate compositional biases. They offer high sensitivity in detecting differential taxa but struggle to control False Discovery Rate (FDR) in sparse microbiome datasets.

LefSe, on the other hand, normalizes count data using TSS or Counts Per Million (CPM), applies nonparametric testing and finally uses Linear Discriminant Analysis to estimate effect size; however, this method is prone to false positives in high dimensional or sparse datasets.

In contrast, methods explicitly designed for compositional data, such as ALDEx2, which employs a Dirichlet multinomial model coupled with Monte Carlo sampling and performs a CLR transformation data and ANCOM-BC2, which uses a log-transformed regression-based approach and adds a regularization step to stabilize variance estimates, enhance the robustness of differential abundance analysis as they reduce false positives while maintain FDR control by accounting for technical and biological variation in compositional data.

Nearing et al. [146] systematically evaluated 14 differential abundance methods across several *16S rRNA* gene datasets, revealing that count-based approaches such as edgeR and limma-voom tend to identify a much larger number of significant taxa compared to more conservative methods. In contrast, ALDEx2 and ANCOM produced fewer significant results, but the taxa they detected were far more reproducible across datasets, suggesting a lower FPR. Furthermore, this author evaluated the discriminatory potential of individual ASVs identified by each tool through ROC curves and the Area Under the Curve (AUC). They observed that ASVs identified by ALDEx2 and ANCOM-II exhibited the highest mean AUROC across datasets, both when using relative abundances and CLR-transformed data, although these methods occasionally failed to detect any significant ASVs even in cases where other tools achieved high AUROCs (e.g., 0.8–0.9). To further evaluate performance, the authors compared precision, recall, and F1-scores of different methods using AUROC thresholds. At a threshold of 0.7, ALDEx2 and ANCOM-II achieved the highest precision values (often approaching 1.0), but this came at the cost of very low recall, especially when contrasted with tools such as LEfSe and edgeR, which displayed much higher recall but lower precision. When CLR data were used, limma-voom and the Wilcoxon test applied to CLR abundances showed among the highest F1-scores, reflecting a better balance between sensitivity and specificity. At a more stringent AUROC threshold of 0.9, most tools showed relatively high recall, but precision declined sharply across all frameworks, consistent with the expectation that few taxa exceed such strong discriminatory performance.

The trade-offs between precision, recall, and F1-score, as highlighted by Nearing et al. [146], emphasize the need for careful tool selection, prevalence filtering, and consideration of both statistical robustness and biological interpretability when drawing conclusions from microbiome differential abundance analyses. More generally, it is advisable to employ multiple methods in parallel and to focus on features consistently identified across tools, while keeping in mind the specific assumptions and limitations inherent to each framework.

#### 3.2.2. Multivariate Analysis

In regression-based analysis, the main aim centers on elucidating relationships between microbial abundances and functional capabilities with specific predictors or outcomes. To this end, researchers typically employ Generalized Linear Models (GLM), primarily those based on Poisson and Negative Binomial distributions to relate features to predictors. However, due to the characteristics of microbiome data such as sparsity and zero inflation, the GLMs assumptions are not met, resulting in biased estimates or unreliable inference [147,148].

To address this situation, a new set regression-based framework has been developed to capture the distributional complexity of microbiome data. MiRKAT-MC [149] is a Kernel and distance-based model that extends association testing to multicategorical outcomes, especially useful for nominal or ordinal phenotypes as they are able to detect global community shifts across disease stages or treatment groups. Some parametric models such as the Zero Inflated Generalized Dirichlet Multinomial (ZIGDM) [150] provide flexible covariance structures and explicit modeling of structural and biological zeros, improving inference on mean shifts and dispersion patterns, aiding in distinguishing heterogeneity in taxa variability that is overlooked by simpler models. Similarly, negative binomial factor regression methods, including reduced-rank (NB-RRR) and co-sparse formulations (NB-FAR) [151], have been developed to uncover latent factors that link host covariates to microbial consortia. Complementing these, adaptive strategies such as the Adaptive Microbiome Association Test (AMAT) combine distance-based feature selection with powerful omnibus testing, making them highly suitable in high-dimensional contexts where signal detection among many irrelevant taxa is a major obstacle [152]. Finally, Bayesian nonparametric formulations, such as zero-inflated multivariate negative binomial regression under dependent Dirichlet process priors, provide highly flexible inference at both the taxon and community levels, disentangling presence/absence from abundance effects; these are especially advantageous for characterizing diversity changes and rare taxa contributions in complex microbiomes [153]. Collectively, these developments mark a clear transition from classical GLMs toward more flexible, high-dimensional regression frameworks specifically adapted to the inherent characteristics of microbiome data, which not only facilitate the identification of disease-associated factors but also enable a rigorous evaluation of their predictive capacity, as exemplified by ROC curves (Figure 3F).

### 3.3. Integration with Multi-Omics Data

Integrating metagenomic data with metatranscriptomic, metaproteomic, and metabolomic analyses provides a comprehensive understanding of microbial communities and their functions. Metagenomics elucidates the genetic potential of these communities, while metatranscriptomics identifies actively expressed genes. Metaproteomics determines the proteins being synthesized, and metabolomics profiles the metabolites present within the system. This integration goes beyond individual layers of analysis, offering insights into the dynamic flow of molecular information from genes to metabolites. By combining these data types, researchers can uncover functional interactions, such as how gene expression drives protein production and metabolite synthesis, linking microbial structure to function. It also facilitates the study of microbial responses to environmental changes, identification of biomarkers, and understanding of host-microbiota interactions [154,155,156].

Careful experimental considerations are crucial for the successful integration of multi-omics analyses. While a detailed exploration of these considerations is beyond the scope of this work, it is important to highlight the need for well-designed study plans, clear definitions of scope and limitations, and a thoughtful selection of omics types that balance information gain, feasibility, and costs. Proper planning ensures statistical power through appropriate sample sizes, controls, replication, and sufficient biomass for multiple assays. Furthermore, attention to the stability of omic analytes and specific collection and preservation conditions is essential to maintain sample integrity. For a deeper understanding, specialized articles on these topics are recommended [156,157,158,159].

Beyond experimental design, multi-omics integration faces important practical challenges. A pervasive issue is missing data, which can arise from multiple sources such as limited sensitivity of analytical platforms, differences in sequencing depth, incomplete overlap of omics assays, poor tissue quality, or stochastic variability in peptide/protein detection. In proteomics, for instance, 20–50% of peptide intensities may be missing across runs, while metabolomics can be affected by instrument-specific biases in ionization and detection. Traditional strategies like complete-case or available-case analyses reduce sample size and often introduce bias, while naïve imputation methods (e.g., mean or zero replacement) distort variance and correlations. To address this, more advanced approaches have been proposed. ML and AI methods such as k-nearest neighbors (KNN) imputation, random forest imputation, and expectation-maximization algorithms better capture relationships among features [160]. Recent transfer-learning approaches, such as TDImpute, use external large-scale datasets (e.g., TCGA) to impute missing values across omics layers by exploiting correlated modalities. Bayesian multi-omics frameworks and tensor decomposition methods further extend these strategies by leveraging shared latent structures across different omics types, improving robustness in the presence of partially observed datasets. These developments highlight that handling missing data is not merely a preprocessing step but a central methodological challenge that directly impacts the validity of integrative analyses [160].

A second major challenge is the correction of batch effects and technical variability, which are ubiquitous in multi-omics studies due to variations in protocols, platforms, and sequencing runs. Batch effects can obscure biological signals and inflate false positives if left uncorrected [159,161,162]. Classical approaches such as ComBat (empirical Bayes framework) and Remove Unwanted Variation (RUV) methods adjust for systematic shifts across batches while preserving biological variance. However, these approaches assume linear batch effects and may underperform in highly heterogeneous multi-omics datasets. To overcome this, newer algorithms integrate batch correction into the latent space modeling of the data, as in MOFA+, which accommodates batch covariates while performing dimensionality reduction [163]. More recently, deep learning–based strategies, including adversarial networks and variational autoencoders, have shown promise in harmonizing multi-omics data by disentangling technical artifacts from biological signals [160]. Ultimately, careful selection and benchmarking of batch correction methods are essential to prevent overcorrection and to ensure that true biological variability is retained.

Different methodological approaches are employed by specialized tools to perform multi-omics integration analysis, each designed to address the specific characteristics of the data and the objectives of the study.

Multi-omics factor analysis (MOFA) [163] uses latent factor analysis to identify patterns of co-variation between features from different omics datasets. This unsupervised approach employs linear models to explain the greatest variability in the data and handles missing values, enabling the integration of partially overlapping datasets. mixOmics [164] includes supervised and unsupervised multivariate analyses. Its DIABLO [165] approach facilitates supervised integration by identifying discriminative features between phenotypes. It assumes linear relationships and requires fully overlapping datasets. MiBiOmics [166] combines multiple methods such as Weighted Gene Correlation Network Analysis (WGCNA), dimensionality reduction (mCIA and Procrustes), and network-based phenotype analyses. This approach allows studying significant interactions across omics layers. COMBI [167] uses latent variable models with log-ratio link functions to handle compositional data, accounting for mean-variance modeling and the relative nature of omics data. It provides joint visualizations to explore associations between features.

Altogether, while tools like MOFA, mixOmics, MiBiOmics, and COMBI have broadened the landscape of integrative analyses, their practical application demands rigorous handling of missing data and robust batch effect correction. Failure to address these issues risks producing spurious associations or losing meaningful biological insights. By integrating advanced imputation methods and harmonization strategies, researchers can improve the reliability and interpretability of multi-omics integration [160,162,168].

### 3.4. Network and Interaction Analysis

Network analysis in metagenomics research has emerged as a key tool to untangle microbial interactions in human microbiomes and their impact on host health conditions, as they allow us to perceive interaction patterns associated with certain diseases and, more recently, metabolic and functional interactions. To this end, computational approaches have been developed to infer and validate relationships between species, offering a wide perspective of the structure and function of complex microbial communities. These approaches rely on statistical methodologies designed to cope with the compositional and sparse nature of the datasets [134,169].

The most widely used strategies can be divided into correlation-based methods and probabilistic graphical models. Correlation-based approaches, such as Sparse Correlation of Compositional Data (SparCC) [170] and CCLasso [171], transform compositional data through log-ratios to estimate pairwise associations. Their main strength lies in computational simplicity and interpretability, but as pointed out in comparative studies, these methods are limited by the assumption of linear correlations and their inability to disentangle direct from indirect associations, which often leads to inflated network connectivity.

In contrast, graphical models such as SParse InversE Covariance Estimation for Ecological Association Inference (SPIEC-EASI) [172] and gCODA [173] have been introduced to estimate conditional dependencies. These models exploit penalized likelihoods and sparse inverse covariance estimation, allowing a more reliable inference of direct interactions. Notably, benchmarking studies highlight that graphical models tend to generate sparser and more biologically plausible networks compared to correlation-based methods, particularly in high-dimensional dataset [174].

Beyond co-occurrence, functional and metabolic network approaches in human metagenomics employ constraint-based models to predict cross-feeding, metabolite fluxes, and the impact of diet or drugs on the gut ecosystem, linking specific taxa to disease-associated shifts in metabolites such as SCFAs, bile acids, or tryptophan derivatives—capabilities that co-occurrence analyses alone cannot provide [175]. By integrating biochemical constraints and flux predictions, these frameworks capture host–microbe exchanges central to human physiology, with perturbations repeatedly linked to obesity, type 2 diabetes, atherosclerosis, and their modulation by interventions like bariatric surgery or exercise [176]. Disease-specific rewiring is evident in inflammatory bowel disease, where depletion of butyrate-producing clostridia and expansion of Enterobacteriaceae feed back on inflammation, while pathobionts such as adherent-invasive *E. coli* exploit L-serine and *Veillonella* spp. switch to nitrate respiration to thrive in the inflamed gut. These mechanistic insights underscore how metabolic networks extend the interpretability of microbiome data, offering avenues for interventions ranging from prebiotics and dietary fibers to rational fecal microbiota transplantation and engineered probiotics, aiming to restore function rather than only composition [177].

Importantly, downstream analyses of these inferred networks provide an additional layer of insight. Network theory offers a suite of topological metrics such as degree, betweenness, modularity, assortativity or nestedness, that help identify keystone taxa that interact with several other taxons or functions as well as the modular organization of communities [174,178]. These properties are linked to ecological and functional implications: for example, higher modularity often confers resilience to the microbial community by containing disturbances within only few taxa, while betweenness highlights taxa that mediate information or metabolite flow across modules, thereby influencing nutrient exchange and cross-feeding relationships. Similarly, assortativity captures whether highly connected taxa preferentially interact with other highly connected taxa, a pattern that can influence robustness by reinforcing clusters or by increasing vulnerability to targeted perturbations in its absence [179]. Beyond structural attributes, these metrics also illuminate functional consequences: for instance, modules can correspond to metabolic guilds involved in short-chain fatty acid production, nitrogen cycling or bile acid metabolism, which may explain shifts in host physiology or disease progression. Recent reviews emphasize that these stability-oriented metrics provide a mechanistic bridge between network architecture and emergent properties such as resilience, productivity, and metabolic complementarity, and are increasingly used to evaluate how microbial communities respond to environmental stress or host-associated disturbances [179,180].

Taken together, correlation-based methods are useful for exploratory analyses, graphical models provide robustness for inferring direct ecological interactions, and functional/metabolic network frameworks extend these analyses to capture the ecological roles and metabolic complementarity within microbiomes. These approaches gain greater power and reliability when applied to larger datasets with an increased number of samples and features, as well as when supported by more robust computational methods. This growing need for richer datasets and sophisticated analyses directly connects with recent advances in sequencing technologies and the expanding availability of metagenomic data, which frame the challenges and opportunities discussed in the following section.

## 4. AI and Machine Learning in Metagenomics

The increasing affordability of sequencing technologies and the maturing understanding of the importance of microbiome roles have led to an increase in data generation within the field of metagenomics [3]. However, as mentioned before, analyzing metagenomic data presents significant challenges due to its complexity. These datasets are often vast and highly multidimensional, with characteristics such as sparsity, compositionality, and a reliance on incomplete or biased reference catalogs [3]. Addressing these challenges requires sophisticated computational tools capable of extracting meaningful insights from the data, which is where AI and ML come into play.

### 4.1. AI/ML Techniques in Microbiome Data Analysis

AI, particularly ML, offers powerful tools to overcome the analytical challenges of metagenomics. By leveraging algorithms capable of recognizing patterns within vast datasets, ML not only facilitates the extraction of meaningful information but also enables predictions and decisions based on these patterns (Mohseni and Ghorbani, 2024) [181].

The field of ML encompasses a wide range of techniques. It can be broadly divided into two categories:

Supervised learning, where a predictive model is trained using labeled datasets. In this approach, both the input features (e.g., microbial abundances) and the desired outcomes (e.g., disease presence) are provided, allowing the model to learn specific relationships and make accurate predictions [182].

Unsupervised learning, which focuses on uncovering patterns and structures in unlabeled datasets. This makes it particularly useful for exploratory analyses, such as grouping similar microbial communities or identifying new microbial traits without prior assumptions [182].

The combination of these techniques allows researchers to tackle insurmountable challenges in microbiome analysis. AI’s ability to uncover hidden patterns in data enables the discovery of relationships and trends that traditional methods might overlook. These models are also adaptive, capable of being refined as new data becomes available, which enhances their utility over time.

In addition to human health applications, these techniques are crucial for understanding environmental microbiomes. For instance, AI-driven models can analyze soil metagenomes to predict nutrient cycling capabilities or detect shifts in microbial communities indicative of environmental stressors, such as pollution or climate change [5].

### 4.2. Current Applications of ML in Metagenomics

ML has transformed the field of metagenomics by enabling researchers to derive actionable insights from vast, complex datasets. While its applications are diverse, they can be broadly categorized into three main areas:

#### 4.2.1. Disease Prediction

ML algorithms have proven invaluable in predicting diseases by analyzing microbial compositions and functional traits. These models help identify associations between microbial communities and conditions such as cancer, metabolic disorders, inflammatory diseases, etc. [183]. For example, models trained on microbiome data have demonstrated success in predicting diseases like colorectal cancer or type 2 diabetes [184]. However, these are just a few examples within a growing body of work leveraging ML to advance diagnostic and prognostic capabilities in clinical settings.

#### 4.2.2. Identification of Microbial Signatures

ML tools are widely used to identify microbial biomarkers linked to specific health conditions or environmental traits. By analyzing patterns in microbiome data, ML algorithms can differentiate microbial communities, providing insights into host-microbe interactions or ecological dynamics. These signatures are instrumental in developing targeted therapies, monitoring environmental changes, or understanding disease mechanisms [182,185].

#### 4.2.3. Functional Trait Prediction

Another critical application is the prediction of functional traits within microbial communities. ML models can infer the roles of microbes based on genetic or metagenomic data, such as predicting antibiotic resistance genes, metabolic pathways, or symbiotic behaviors. These insights are essential for both medical research and ecological studies, supporting advancements in personalized medicine, biotechnology, and environmental monitoring [183].

While the examples above illustrate the potential of ML in metagenomics, they represent only a fraction of the applications currently in use, underscoring the versatility and growing impact of ML in this field.

### 4.3. AI Tools for Metagenomics

Numerous AI-powered tools have been developed to address specific challenges in metagenomics, ranging from disease prediction to functional annotation. Some notable tools are mentioned in Table 5.

Selected tools and their applications, highlighting key advancements in disease prediction, functional annotation, and microbial feature identification.

### 4.4. Comparison with Traditional Tools

Traditional metagenomics analysis relies heavily on sequence alignment and database searches, which, although effective, have notable limitations [3,191,192]:Computational resources: Traditional methods often require significant computational resources, particularly for large datasets.Limited Novelty Detection: Identifying previously uncharacterized microorganisms or genes is challenging using traditional approaches that depend on existing databases. AI-based tools address these challenges by offering enhanced capabilities [191,192].Efficiency in Application: The computational profile of AI-based tools presents a critical trade-off between training and application. While the process of training deep learning models is often famously resource-intensive, requiring large datasets and significant computational power, the resulting models can be highly efficient for inference. Once trained, applying a model to classify new sequences or predict a phenotype is often computationally much faster than performing traditional, per-sample alignment-based searches against large reference databases [3].Pattern Recognition: Advanced algorithms identify subtle patterns and relationships that traditional tools may overlook, such as novel microbial signatures or complex microbial interactions.Scalability: AI tools are well-suited for managing the ever-expanding scale of metagenomic data, enabling large-scale analyses that were previously impractical.

By combining AI’s predictive power with traditional methods’ robustness, researchers can achieve a more comprehensive understanding of microbial communities, unlocking new opportunities in metagenomics research.

### 4.5. Future Directions: Advancing AI in Metagenomics

As AI continues to evolve, its integration with microbiome analysis is expected to unlock new avenues for individualized healthcare and address longstanding challenges in data processing and interpretation.

#### 4.5.1. Enhancing Personalized Medicine

AI has the potential to revolutionize personalized medicine by leveraging insights derived from an individual’s microbiome composition [5].

Disease Susceptibility and Treatment Response: AI models can analyze an individual’s microbiome to predict their susceptibility to specific diseases, as well as their likely response to treatments. By identifying microbial patterns associated with these conditions, AI-driven predictions could facilitate earlier diagnoses and improve therapeutic outcomes [3,183].

Personalized Interventions: These predictions can inform interventions, including customized dietary recommendations, probiotics, or microbiome-targeted therapies, tailored to an individual’s unique microbial composition. For example, ML models could optimize dietary plans by identifying foods that enhance beneficial microbial functions or mitigate dysbiosis [181].

#### 4.5.2. Overcoming Challenges: Benchmarking, Robustness, and Interpretability

Despite the promising potential of AI in metagenomics, several critical challenges must be addressed to fully realize its benefits [3]. In this section we emphasize three practical and interrelated requirements to translate methodological promise into reliable, translational results.

##### Benchmarking and External Validation

Many published tools—including DeepARG, Meta-Signer, and mAML—show strong performance on internal training cohorts but have limited assessment on independent datasets [186,188]. Meta-analyses and cross-cohort studies have repeatedly shown that microbiome classifiers often suffer performance drops when moved across populations and technical settings, highlighting the need for cross-region/cross-cohort validation [193,194]. We therefore recommend that studies report external validation on hold-out cohorts (ideally collected at different sites), and adopt community benchmarks or meta-analysis toolboxes (e.g., SIAMCAT) to quantify generalizability across datasets [195]. Where external cohorts are unavailable, nested cross-validation with fully separated tuning and testing folds should be used to avoid optimistic bias [196].

##### Overfitting Mitigation and Uncertainty Estimation

High dimensionality, compositional constraints, and small sample sizes commonly exacerbate overfitting in microbiome ML. Best practices include careful feature selection, regularization, ensemble approaches, transfer learning where appropriate, and realistic data augmentation [196]. In addition, model uncertainty should be reported (e.g., via MC-dropout or other approximate Bayesian approaches) to distinguish confident from uncertain predictions and to improve risk assessment for translational use [197]. We also encourage reporting calibration metrics and class-imbalanced performance measures (precision–recall curves, calibration plots) in addition to AUROC.

##### Interpretability, Feature Stability and Biological Validation

Many advanced AI models—particularly deep learning architectures—still behave as “black boxes,” which limits transparency and reduces clinical trust and uptake. To address this, researchers should prioritize explainability both procedurally and architecturally: apply model-agnostic explanation tools while also evaluating inherently interpretable or “interpretable-by-design” architectures where feasible [198,199,200]. Importantly, explanations must be tested for stability (do the same features recur under resampling, different preprocessing choices, or across cohorts?) because feature importance is highly sensitive to data transformation and model selection in microbiome datasets [201,202]. We therefore recommend a reproducibility-oriented workflow for interpretation: use multiple explanation methods (global + local), quantify feature stability with resampling/bootstrap or stability-focused feature-selection methods, perform leave-one-study-out or cross-cohort checks to assess concordance of important features, and whenever possible, seek orthogonal biological validation (e.g., independent cohorts or functional assays) to move from correlation toward mechanistic plausibility [193,203].

Together, these practices reduce false leads from spurious, dataset-specific signals and increase confidence that identified microbial drivers are robust, interpretable, and translationally relevant.

##### Bridging the Gap: Multidisciplinary Collaboration and Innovation

The future of AI in metagenomics depends on fostering collaborations between computer scientists, biologists, and clinicians. Such interdisciplinary efforts are crucial for designing AI models that address real-world challenges in microbiome research. Moreover, integrating AI with emerging technologies, such as single-cell metagenomics and metatranscriptomics, will enable deeper insights into microbial ecosystems and their interactions with hosts [181].

By addressing these challenges and continuing to innovate, AI-driven metagenomics has the potential to revolutionize our understanding of microbial communities, offering breakthroughs in healthcare, environmental science, and biotechnology. The journey ahead promises to not only enhance our ability to combat diseases but also to harness microbial diversity for sustainable solutions to global challenges.

To provide an integrative perspective within this review, the following section aims to share an applied overview on how computational metagenomics tools have been leveraged in order to respond to different biological questions. By outlining this continuum, we intend to illustrate not only the versatility of metagenomic approaches but also their capacity to bridge fundamental ecological systems.

## 5. Applications in Human and Environmental Health

The human microbiome is composed of diverse microbial communities, including bacteria, archaea, viruses, phages, and fungi, which resides in different body sites such as the oral cavity, skin surface, intestinal tract, esophagus, lungs, and more [204]. Microbial colonization is not uniform across the body; each body compartment contains its own microbiota, and even within the same site, microbial composition can vary depending on the specific area of sampling [205].

Microbes interact with one another both within the same species and across different species, genera, families, and even domains of life. These symbiotic relationships influence microbial fitness, population dynamics, and functional capacities. Interactions can be beneficial (mutualism, synergism, or commensalism), detrimental (amensalism, including predation, parasitism, antagonism, or competition), or neutral, where no significant effect on microbial function is observed [205].

Microbial colonization begins at birth and is influenced by factors such as the mode of delivery [206,207] and feeding practices [208]. The microbiome is dynamic and shaped by various elements, including aging [209], diet [210], antibiotic exposure [211,212], disease [213,214], and other environmental influences.

Beyond microbial interactions, the microbiome plays a crucial role in maintaining host health by regulating metabolism, immunity, and homeostasis. Several key functions have been attributed to the core microbiome, including polysaccharide digestion, immune system development, protection against infections, vitamin synthesis, fat storage, angiogenesis regulation, and even behavioral development [215].

However, when dysbiosis occurs—characterized by shifts in microbial composition, reduced diversity, and functional imbalances—these once-beneficial interactions can contribute to the onset of various diseases. Metagenomic approaches have been indispensable in moving beyond simple associations to dissecting these complex interactions.

### 5.1. Microbiome and Disease

Computational metagenomics and microbiome analysis have been applied in a variety of studies to explore the relationship between microbial communities and disease development. Early studies, primarily employing targeted *16S rRNA* gene sequencing, were instrumental in characterizing broad shifts in the taxonomic composition of microbial communities in various tissues and fluids. While foundational, this approach is limited to taxonomic profiling and often cannot resolve below the genus level. The advent of shotgun metagenomics provided a much deeper view by enabling not only higher-resolution taxonomic classification but also functional profiling of the entire gene content of a community and the reconstruction of MAGs. More recently, the emergence of LRS is further revolutionizing the field by allowing for the assembly of complete, closed genomes directly from complex samples, which is critical for strain-level resolution and understanding the genomic context of functional genes [36,216].

#### 5.1.1. Metabolic Diseases

The gut microbiota has been suggested to play an important role in obesity, potentially influencing energy absorption, central appetite regulation, fat storage, chronic inflammation, and circadian rhythms [217]. Although the Firmicutes/Bacteroidetes ratio was initially proposed as a hallmark of obesity [218,219], subsequent meta-analyses and large-scale studies failed to replicate this association consistently. The inconsistency of this finding across populations highlighted the limitations of low-resolution taxonomic markers. The transition to shotgun metagenomics shifted the focus from broad phylum-level changes to the functional capacity of the microbiome. These studies revealed that obesity is more consistently associated with a reduction in overall microbial gene richness and functional diversity, particularly in pathways related to butyrate production and carbohydrate metabolism. By reconstructing MAGs from obese and lean individuals, researchers have been able to link specific microbial lineages, such as members of the Christensenellaceae family, to lean phenotypes [220,221,222,223].

While these short-read shotgun studies provide powerful functional insights, they often yield fragmented MAGs. The use of LRS is now helping to overcome this, making it possible to assemble more complete genomes of key metabolic players like *Akkermansia muciniphila*, which allows for strain-level analysis of its role in metabolic health [36]. This progression demonstrates how advancements in metagenomic methods have enabled a more nuanced understanding of the microbiome’s role in metabolic disease, moving from simple taxonomic ratios to detailed functional and strain-level insights.

#### 5.1.2. Liver Diseases

Some evidence suggests a strong association between gut dysbiosis and the onset and progression of metabolic dysfunction-associated steatotic liver disease (MASLD) [224]. Shotgun metagenomic studies have been crucial in identifying these links. For instance, by comparing the functional gene content of patients with MASLD to healthy controls, researchers have identified an overabundance of pathways related to inflammation and lipogenesis, often linked to genera like Escherichia and Streptococcus. The ability to perform whole-genome assembly from metagenomic data has also allowed for the reconstruction of MAGs of these pathogenic bacteria, revealing virulence factors that may contribute to liver inflammation. Research by Solé et al. [225] used quantitative shotgun metagenomics to demonstrate a clear reduction in gene richness as cirrhosis progresses, directly linking the functional capacity of the microbiome to the severity of liver dysfunction. The next frontier in this area is using LRS to assemble complete viral genomes (the virome), as bacteriophages can modulate bacterial populations and may play a key role in the progression of liver disease [216].

#### 5.1.3. Pregnancy and Reproductive Disorders

Recent research describes the role of the gut microbiome in pregnancy, particularly its influence on maternal and fetal health outcomes. Early investigations using *16S rRNA* sequencing established correlations between broad shifts in the vaginal microbiota, such as a decrease in Lactobacillus dominance, and adverse pregnancy outcomes. However, shotgun metagenomics has been critical in moving beyond taxonomic associations to understanding functional potential [226]. By analyzing the collective gene content, researchers have been able to link changes in metabolic pathways—such as amino acid and carbohydrate metabolism—to conditions like preeclampsia and gestational diabetes mellitus [216].

Furthermore, studies [227,228] have shown that dysbiosis of gut microbiota in women could be associated with Polycystic Ovary Syndrome (PCOS). Shotgun metagenomics has revealed not just a change in species, but a shift in functional capacity toward inflammatory pathways. A key challenge in this area is the high degree of strain-level diversity among critical commensals like *Lactobacillus*. Different strains of the same species can have vastly different probiotic or pathogenic potentials. Therefore, modern approaches using LRS to assemble complete genomes and achieve true strain-level resolution are becoming essential for determining which specific lineages are protective versus detrimental during pregnancy [8]. This highlights a critical application where moving from species-level profiling to high-resolution strain identification is necessary to translate research findings into clinical diagnostics or therapies.

#### 5.1.4. Neurological Disorders

The microbiome also plays a significant role in neurodevelopmental disorders and behavior. The gut–brain axis is one of the most exciting frontiers in microbiome research, and metagenomics provides the primary toolkit for its exploration. Alterations in the gut microbiota have been observed in individuals with autism spectrum disorder (ASD) and other neurodevelopmental disorders. Again, initial findings from *16S rRNA* profiling identified taxonomic markers, such as an increased abundance of Clostridium species in some ASD cohorts. However, the mechanism remained elusive [229,230,231].

The adoption of shotgun metagenomics provided the crucial next step by linking these taxonomic shifts to functional changes. By analyzing the microbial gene content, studies have identified depletions in pathways for synthesizing key neurotransmitters (for example, GABA and serotonin precursors) and short-chain fatty acids (SCFAs) like butyrate, which are known to have neuroprotective effects. To understand if these functional deficits are due to the absence of genes or simply a lack of their expression, researchers are increasingly turning to metatranscriptomics. This approach directly measures microbial gene expression (RNA), revealing which metabolic pathways are actively transcribed by the gut community. This allows a distinction between the genetic potential (DNA) and the real-time functional activity (RNA), which is critical for understanding dynamic conditions like ASD [232]. A comprehensive review by Liu et al. [233] summarizes how these multi-omic metagenomic approaches are building a body of evidence linking the gut microbiome to a wide range of neurodegenerative diseases.

#### 5.1.5. Inflammatory Skin Disorders

In dermatology, recent studies have linked dysbiosis of the cutaneous and intestinal microbiomes with skin-associated diseases, such as psoriasis [234], atopic dermatitis [235], acne vulgaris, rosacea, alopecia areata, and hidradenitis suppurativa [236]. Specifically, a systematic review by Widhiati et al. [237] explored the associations between the gut microbiome with inflammatory skin disorders, suggesting that *Bifidobacterium* plays an essential role as anti-inflammation bacteria, and Proteobacteria and Enterobacteria impact inflammation in inflammatory skin disorders.

The reconstruction of these MAGs has been vital for identifying strain-specific virulence factors and metabolic functions that correlate with disease states like acne or atopic dermatitis. Moreover, metatranscriptomics is being used to determine which microbial pathways are active on the skin during inflammatory flares versus periods of health. By sequencing the expressed RNA, it is possible to identify if, for example, certain strains of *Staphylococcus aureus* are actively transcribing toxins or if commensal microbes are expressing anti-inflammatory molecules [232]. This functional insight provides a much more dynamic picture of the skin microbiome’s role in disease than what can be gleaned from DNA-level surveys alone.

#### 5.1.6. Autoimmune Diseases

Inflammatory bowel disease (IBD), a chronic relapsing-remitting disorder of the gastrointestinal tract, is closely linked to intestinal microbiota. IBD represents a paradigm for the power of advanced metagenomic analysis. The field has progressed significantly from early *16S rRNA* studies, which first described a general state of dysbiosis in IBD, characterized by a loss of butyrate-producing bacteria like *Faecalibacterium prausnitzii*. While important, these studies could not explain the functional consequences of this dysbiosis [238].

The breakthrough came with the application of integrated, multi-omic approaches, exemplified by large-scale longitudinal studies like the Integrative HMP [239]. By combining shotgun metagenomics (MGX), metatranscriptomics (MTX), metaproteomics (MPX), and metabolomics, this study provided a holistic view of the IBD gut ecosystem.
Metagenomics confirmed the depletion of key commensals and provided the genomic blueprints (via MAGs) of the community members.Metatranscriptomics revealed that during disease flares, the remaining microbes, including pathobionts like *Ruminococcus gnavus*, were transcriptionally hyperactive, indicating a highly stressed and inflammatory environment [232,239].Metaproteomics complemented this by providing direct evidence of which proteins—the functional workhorses of the cell—were being produced, confirming that inflammatory and stress-response pathways were highly active [240].Metabolomics measured the downstream biochemical output, identifying clear shifts in bile acid metabolism and SCFA production that correlated with the functional changes observed at the gene and protein level.

This integrated metagenomic approach revealed that IBD is not just a change in who is there, but a fundamental rewiring of the entire molecular activity of the microbiome. It is the ability to combine these different layers of metagenomic data that is leading to new diagnostic biomarkers and therapeutic targets for IBD.

### 5.2. Therapeutic Applications

The clinical application of microbiome-targeted therapies relies heavily on metagenomic tools for both development and monitoring.

#### 5.2.1. Probiotics

Probiotics are live microorganisms that, when administered in adequate amounts, confer a health benefit on the host [241]. The effectiveness of probiotics is highly strain-specific. Therefore, simply using *16S rRNA* sequencing to confirm the presence of a *Lactobacillus* species is insufficient. Shotgun metagenomics, and particularly LRS is required for the high-resolution tracking of specific probiotic strains to confirm their engraftment and functional activity in the gut [8].

Some of the species from which strains with probiotic characteristics have been isolated are *Lacticaseibacillus casei*, *Lacticaseibacillus paracasei*, *Lacticaseibacillus rhamnosus*, *Lactiplantibacillus plantarum*, *Lactobacillus acidophilus*, *Levilactobacillus brevis*, *Ligilactobacillus salivarius* [242].

These are commonly found in probiotic formulations, such as dietary supplements or medicine, and in foods like fermented dairy products or other fermented foods [242].

According to Wang et al. [243], probiotics play a significant role in maintaining and modulating gut microbiota composition. They can adjust the structure of human intestinal microorganisms and inhibit the colonization of pathogenic bacteria in the intestine. This inhibition occurs through various mechanisms, including stimulation of epithelial barrier function, production of antimicrobial substances, restriction of pathogenic access to nutrient resources, and competition for binding sites. Additionally, probiotics secrete organic acids, such as butyric acid, acetic acid, and propionic acid, during carbohydrate fermentation. These acids lower the intestinal pH, thereby creating an environment that inhibits the growth of harmful bacteria.

Probiotics also contribute to intestinal health by enhancing the protective layer of intestinal mucosa, thereby strengthening the intestinal barrier and improving immune function. Notably, butyric acid, a key metabolite produced by probiotics, promotes oxygen consumption in the intestinal epithelium. This process increases the expression of barrier-protective hypoxia-inducible factor (HIF) target genes and helps maintain HIF stability, which is essential for intestinal homeostasis [243].

Furthermore, probiotics enhance immune responses by increasing the number and functionality of macrophages and dendritic cells (DCs) in the lamina propria. They also promote the maturation of humoral immunity by stimulating the production of immunoglobulin A (IgA) antibodies, which play a critical role in mucosal defense [243].

Given their ability to modulate gut microbiota, probiotics have emerged as a promising therapeutic approach for managing various microbiota-related diseases. Evidence suggests that certain probiotic strains are effective in preventing antibiotic-associated diarrhea in both adult and pediatric populations. Acute gastroenteritis remains the original and probably the most well-established clinical indication for probiotic use [244]. However, there are still issues with dose and timing that require further research.

#### 5.2.2. Prebiotics

A prebiotic is a substrate that is selectively utilized by host microorganisms, conferring a health benefit [245]. Among the most studied prebiotics are oligosaccharide carbohydrates, of which mainly include xylooligosaccharides (XOS), galacto-oligosaccharides (GOS), lactulose and inulin and its derived fructose-oligosaccharides [246]. These compounds are selectively fermented in the colon, leading to the production of short chain fatty acids (SCFAs) and biomass, decreasing the pH value of the intestinal environment. Therefore, they create favorable conditions for the growth and activity of beneficial bacteria, such as bifidobacteria and lactobacilli (e.g., *L. plantarum*, *L. paracasei*, *Bifidobacterium bifidum*), while simultaneously inhibiting the proliferation of pathogens (*Clostridium perfringens*, *Escherichia coli*, *Campylobacter jejuni*, *Enterobacterium spec.*, *Salmonella enteritidis* or *Salmonella typhimurium*), thereby contributing to improved human health [247].

Prebiotics occur naturally in a variety of foods, including wheat, onions, bananas, honey, garlic, and leeks [242]. Their health benefits include improving the regulation of immunity, resisting pathogens, influencing metabolism, increasing mineral absorption, and enhancing health [246].

#### 5.2.3. Fecal Microbiota Transplantation

Fecal microbiota transplantation (FMT) is a treatment that involves the administration of a minimally manipulated microbial community from the stool of a healthy donor into a patient’s intestinal tract to restore normal gut microbiota function [248].

FMT is an important therapeutic option for *Clostridioides difficile* infection (CDI). Its use in treating recurrent CDI has been shown to be well tolerated and effective when primary treatment has failed [249].

Metagenomics is the primary tool for assessing the success of FMT. Shotgun sequencing is used to perform deep taxonomic and functional profiling of both the donor’s and recipient’s microbiota. Post-transplantation, this analysis is repeated to track the engraftment of donor strains and the restoration of key metabolic pathways that were deficient in the recipient. This provides a quantitative, data-driven measure of therapeutic success [250].

#### 5.2.4. Precision Medicine

As mentioned previously, the function, composition, and growth dynamics of the gut microbiome are associated with many host physiological and pathological states. This evidence highlights the potential of the gut microbiome in precision medicine and individualized treatment.

Non-invasive sampling methods and decreasing profiling costs make it a feasible tool for early diagnosis and disease risk assessment. Integrating features related to the composition of the gut microbiome with other known clinical risk factors may potentially enhance early disease detection [251].

This will be driven by the integration of multi-omics data, similar to what has been done for IBD [239]. By generating a comprehensive metagenomic, metatranscriptomic, and metabolomic profile of a patient, it may become possible to computationally predict their response to a specific diet, probiotic, or therapeutic drug. The development of robust, high-throughput computational pipelines for integrating these complex datasets is therefore a critical area of ongoing research that will be essential for bringing precision microbiome medicine into the clinic [216].

However, many challenges and limitations still need to be addressed for microbiome profiling to be fully integrated into common medical practice. Zmora et al. [251] state that the microbiome shows a remarkable degree of inter-personal variability, with even intra-personal fluctuations that tend to oscillate diurnally. Additionally, differences in sample collection and analysis techniques, reagents, and parameters may introduce variations into microbiome results, potentially adding biases to the interpretation of microbiome-based data.

The field must develop robust and standardized protocols for sample collection, sequencing, and analysis to improve the reproducibility of results and reduce biases.

## 6. Data Sharing and Open Science

In the last decade enormous quantities of metagenomic data have been generated due to the reduction in costs and the accessibility of high-throughput sequencing techniques. Although most of this data has been deposited in different repositories such as the NCBI SRA [106], the ENA [252], or the EMBL-EBI [253], a concerning number of studies that lack proper data annotation and metadata related to the sample, restricting the re-analysis of the data for further and larger studies and leading the set of microbiome data to being “single-use” without any further repurpose [254,255]. The FAIR principles are the gold standard for the administration and management of data

To address these challenges, the FAIR (Findability, Accessibility, Interoperability and Reusability) principles [256] have emerged as a gold standard framework to enhance the usability and longevity of diverse scientific data as they advocate for the assignment of unique identifiers for each dataset, provision of rich metadata for the reproducibility of the analysis and the use of standardized vocabulary to standardize the bulk of data in public for effective retrospective studies. Along with this principles, the Genomic Standards Consortium implemented the Minimal information for Marker Genes Sequences (MIMARKS) and the Minimal Information about any Sequence (MIxS) [257] in 2011 and in 2012 the MINSEQE [258] guidelines to keep track of key information regarding the description of the biological system and condition studied, as well as the sequence read data, the final processed data from the assays in the study and essential metadata with processing protocols.

While the FAIR principles provide a valuable framework for data stewardship, their real-world application in human microbiome research encounters several hurdles:

Metadata Heterogeneity: Studies vary widely in how they record sample source, collection protocols, sequencing methods, and participant demographics. This lack of consistency complicates meta-analyses and cross-study comparisons [254,255]. To mitigate this, we recommend adopting universal minimal metadata checklists—such as the MIxS standards—but extending them with controlled vocabularies for host characteristics and clinical covariates.

Repository Limitations: Public archives like NCBI SRA, ENA, and EMBL-EBI differ in submission interfaces, metadata validators, and update policies, leading to incomplete or outdated records [106,252]. We suggest that consortia develop centralized dashboards to monitor submission completeness and automatically flag missing fields or format mismatches.

Legal and Ethical Frameworks: Human microbiome datasets inevitably co-sequence trace amounts of host DNA, raising re-identification risks when coupled with detailed metadata [259]. Currently, few jurisdictions provide clear regulations on sharing de-identified human microbiome data. We call for international guidelines that balance open science with privacy—potentially modeled on GDPR for genomic data—and for the use of cryptographic or synthetic-data approaches to allow sharing of derived features without exposing raw sequences.

Anonymization and Controlled Access: Beyond technical anonymization (e.g., removing direct identifiers), controlled-access repositories (e.g., dbGaP, EGA) can provide tiered data-sharing mechanisms. We recommend implementing data-use agreements that specify acceptable secondary analyses and require proof of IRB approval.

By addressing these challenges through community-driven standards, improved repository infrastructure, and robust legal safeguards, the field can move toward truly FAIR and ethically responsible microbiome research.

## 7. Next Steps in Metagenomics

Recent methodological advances are reshaping the scope of metagenomic research by addressing longstanding limitations in assembly, taxonomic resolution, and the representation of microbial variability. Single-cell metagenomics has emerged as a powerful complement to metagenome MAGs, enabling the recovery of Single Amplified Genomes (SAGs) from low abundance and rare taxa. By bypassing the reliance on binning and assembly, this strategy reduces chimerism and enhances resolution at strain level, allowing the accurate reconstruction of microbial populations and gene content [260,261]. Similarly, pangenome graphs are redefining reference frameworks by moving beyond linear genome representations. Instead of relying on a single reference, graph-based models incorporate genetic variability such as single-nucleotide variants, indels and structural rearrangements, providing a more nuanced view of intra-species diversity and enabling high resolution classification of metagenomic reads at the strain level [262]. These developments expand the analytical toolbox available for metagenomics and facilitate the exploration of microbial diversity with unprecedented precision.

The translation of these advances into clinical and epidemiological practice has become evident. Real-time nanopore sequencing has demonstrated its utility in the rapid diagnosis of infections directly from patient samples. In cerebrospinal fluid, metagenomic nanopore sequencing outperformed conventional culture-based approaches, enabling pathogen identification within minutes and proving particularly useful in cases where standard diagnostic assays fail [263]. Beyond diagnosis, the capacity of LRS platforms to perform on-site analysis has been highlighted as a key factor in outbreak management and epidemiological surveillance. Their ability to detect structural variants, plasmids and mobile genetic elements in near real time enhances our capacity to track antimicrobial resistance and pathogen evolution during public emergencies [264]. In parallel, single-cell metagenomics offers a route to link specific antibiotic resistance genes or metabolic functions directly to their microbial hosts, a level of resolution with clear clinical relevance for antimicrobial stewardship and personalized microbiome targeted interventions [260].

Despite these promising advances, significant challenges remain before these approaches can be broadly implemented. Single-cell metagenomics is limited by technical hurdles such as contamination, amplification bias, and uneven genome coverage, which can compromise the quality of SAGs [261]. Moreover, while single-cell workflows are becoming more scalable, they remain resource-intensive compared to bulk metagenomic sequencing. Nanopore sequencing, although invaluable for rapid diagnostics, still suffers from relatively high error rates compared to short-read technologies, which can complicate the detection of low-frequency variants or subtle strain differences [263]. Similarly, while pangenome graphs provide a conceptually robust framework, they require extensive computational resources and high-quality reference collections to build and curate graph structures. Their adoption also depends on the development of standardized methods and visualization tools that make graph-based analyses accessible to non-specialist users [262]. Taken together, these limitations emphasize that while single-cell approaches, real-time sequencing, and pangenome frameworks represent the next frontier in metagenomics, further refinement and validation are essential for their integration into routine research and clinical pipelines.

## Figures and Tables

**Figure 1 ijms-26-09206-f001:**
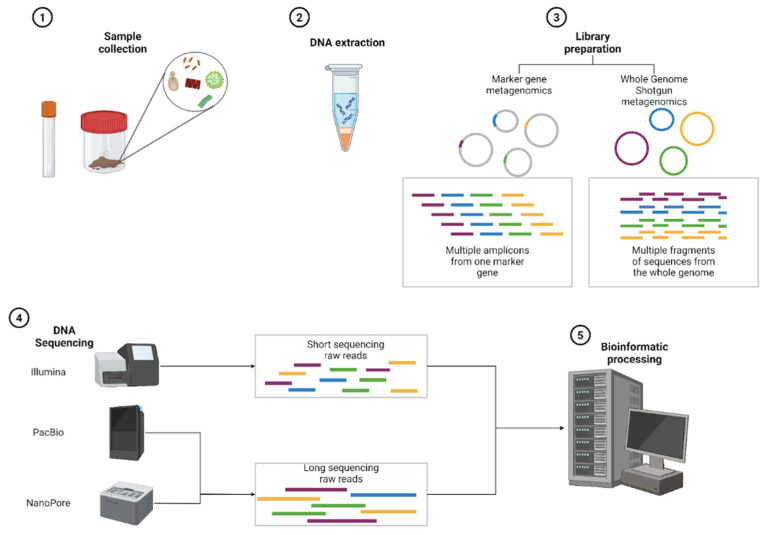
Schematic representation of the wet lab workflow in metagenomic studies. The biological question and study design are critical as all subsequent methodologies and approaches depend on them. (1) Representative samples are collected from the microbiome of interest. (2) Genetic material (DNA or RNA) is extracted using an appropriate extraction kit. (3) Marker genes or whole genome libraries are prepared. (4) The libraries are sequenced, selecting the appropriate sequencing platform and format. (5) The resulting raw sequencing data is processed following bioinformatic pipelines workflows (figure made with Biorender.com (Created in BioRender. Hernandez-Lemus, E. (2025) https://BioRender.com/ozn2z0p).

**Figure 2 ijms-26-09206-f002:**
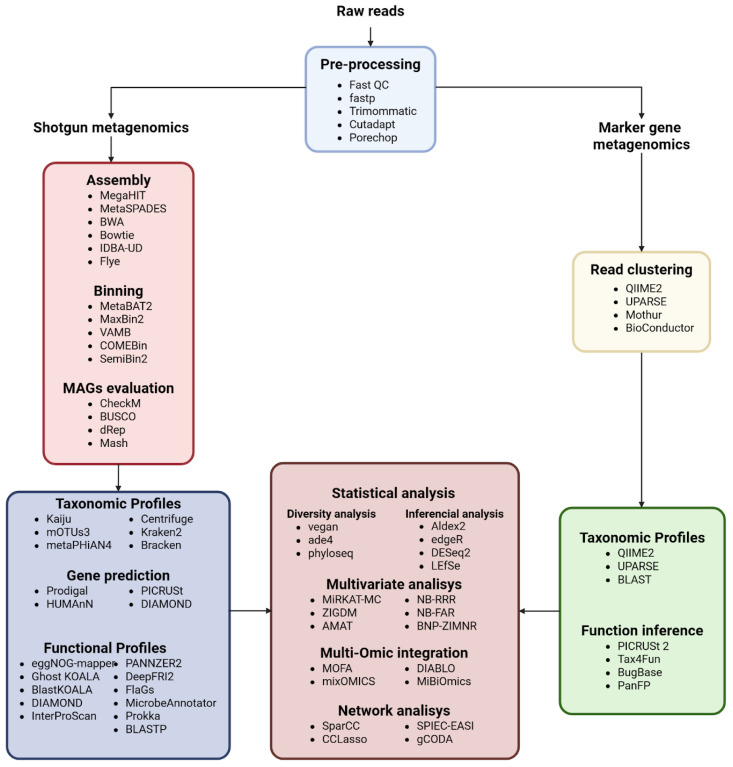
Standard workflow and associated bioinformatic software employed for shotgun and marker genes metagenomics (Created in BioRender. Hernandez-Lemus, E. (2025) https://BioRender.com/igjz7bp).

**Figure 3 ijms-26-09206-f003:**
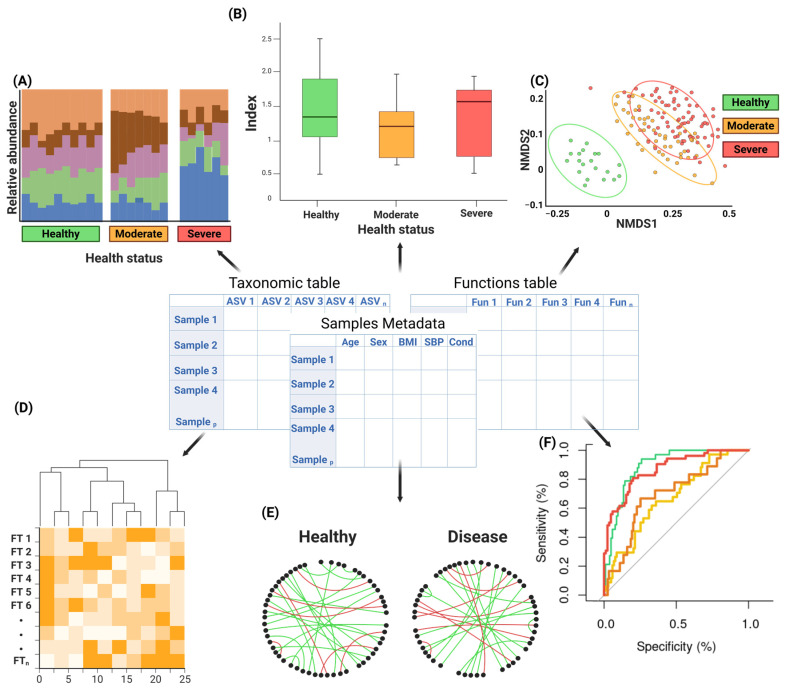
Visualization methods for downstream analysis output representing (**A**) relative abundance graphics, (**B**) alpha diversity boxplots, (**C**) Ordination plots for beta diversity, (**D**) heatmap for differential abundance, (**E**) network analysis and (**F**) receiver operating characteristic (ROC) curve (Created in BioRender. Hernandez-Lemus, E. (2025) https://BioRender.com/ltrclbu).

**Table 1 ijms-26-09206-t001:** **Sequencing Technology Comparison for Metagenomic Applications.** * Species-level resolution variable depending on taxa and database completeness ** Strain-level resolution requires high depth and/or assembly-based approaches.

Technology	Taxonomic Resolution	Typical Input DNA	Typical Sequencing Depth	Main Strengths	Primary Limitations
*16S rRNA* (Illumina)	Genus → Species *	5–50 ng	10–100 K reads/sample	Standardized workflows, extensive databases, cost-effective	Limited to prokaryotes, poor species resolution, no functional data
Shotgun SRS (Illumina)	Species → Strain **	100–500 ng	5–50 M reads/sample	High accuracy, established tools, functional profiling	Poor repetitive region assembly, reference-dependent
PacBio HiFi	Species → Strain	1–10 μg HMW DNA	0.5–5 M reads/sample	Very high accuracy, excellent MAG recovery, complete genomes	Higher cost, shorter reads than ONT
ONT MinION/GridION	Species → Strain	1–10 μg HMW DNA	0.5–5 M reads/sample	Ultra-long reads, real-time sequencing, structural variants	Moderate accuracy in homopolymers, specialized workflows

**Table 2 ijms-26-09206-t002:** Comparison of contig/genome assembly tools.

Tool	Approach	Advantages	Limitations
MEGAHIT	De Bruijn Graph Ultra-fast assembly	Extremely low memory consumption; suitable for large complex metagenomes	May yield lower completeness in complex communities; shorter contigs compared to metaSPAdes
metaSPAdes	De Bruijn Graph Produces longer contigs	High fraction of reads assembled; optimized for metagenomic data	Higher mis-assembly rate; more resource intensive; longer runtime
BWA/Bowtie2	Reference-guided alignment	High accuracy for known taxa; improved resolution for well-characterized genomes	Not suitable for novel or poorly represented organisms; requires high-quality reference databases
IDBA-UD	Iterative De Bruijn Graph	Handles uneven sequencing depths well; good for complex communities	Slower than MEGAHIT; higher memory requirements than succinct graph approaches
Flye	Overlap-layout-consensus (long-read)	Excellent for long-read data; produces highly contiguous assemblies; good error correction	Primarily designed for long reads; computationally intensive for large datasets

**Table 3 ijms-26-09206-t003:** Comparison of metagenomic binning tools.

Tool	Approach	Advantages	Limitations
MetaBAT2	Hybrid clustering using coverage and k-mer profiles	Widely used and well-validated; effective use of abundance patterns; good performance on diverse datasets	Requires contigs ≥ 1500 bp for optimal performance; struggles with low-abundance species
MaxBin2	Expectation-maximization with abundance and composition	Effective for low-abundance species; robust statistical framework; handles uneven coverage well	Less effective on very complex communities; sensitive to parameter settings
VAMB	Deep learning with variational autoencoder	Robust across different contig length thresholds; learns complex feature representations; good scalability	Requires substantial training data; black-box approach limits interpretability
COMEBin	Contrastive multi-view representation learning with Leiden clustering	Demonstrates higher recovery of near-complete bins; uses advanced community detection; data augmentation improves robustness	Emerging method with limited field validation; computationally intensive; requires expertise to optimize
SemiBin2	Semi-supervised deep learning with Siamese networks	Leverages both labeled and unlabeled data; good performance across diverse environments; user-friendly	Requires some reference genomes for training; newer tool with growing but limited validation

**Table 4 ijms-26-09206-t004:** Taxonomic profiling and functional annotation tools that have been benchmarked previously.

Process	Tool	Dataset	Memory Use	Running Time	Source
Taxonomic profiling	UPARSE	4.7 Million sequences	Not specified	1 h	Marizzoni et al., 2020 [84]
	Bioconductor			~8 h	
	QIIME2			3 h	
	Mothur			9 h	
	mOTUs 3	50 randomly selected human gut metagenomes	~15 GB	~4 min	Shaw & Yu, 2025 [85]
	MetaPhIAn 4		~18 GB	~6 min	
	Centrifuge	5.7 million sequences	20 Gb	7 min	Ye et al., 2019a [86]
	Kraken2		36 GB	1 min	
	Bracken		<1 GB	<1 min	
	CLARK		80 GB	2 min	
Functional Profiles	PICRUSt2	Human HMP dataset (Not specified)	~15 GB	~45 min	Mongad et al., 2021 [63]
	MicFunPred		~6 GB	~30 min	
	Tax4Fun2		~4 GB	~15 min	
	eggNOG-mapper v2	4296 coding sequences for *Escherichia coli K-12*	6 GB	~26 min	Alonso-Reyes & Albarracin, 2024 [87]
	GhostKOALA	3.8 Million sequences	Web service	~6 min	Kanehisa et al., 2016 [71]
	BlastKOALA	2.6 Million sequences	Web service	~41 min	
	DIAMOND	~1.7 million sequences	~14 GB	8 min	Buchfink et al., 2021 [72]
	MMSeqs2		11 GB	53 min	
	BLASTP		Web service	46 days	
	MicrobeAnnotator	100 *E. coli* genomes	~19.4 GB/genome	3.7 h/genome	Ruiz-Perez et al., 2021 [70]
	InterProScan		~4 GB/genome	~2.7 min/genome	
	Prokka		204 MB/genome	~2.5 min/genome	

**Table 5 ijms-26-09206-t005:** Overview of AI/ML tools for metagenomics analysis.

AI/ML Tool	Application
Meta-Signer (Reiman et al., 2021 [186])	Feature ranking through ensemble learning and metagenome signature identification
DeepMicro (Oh & Zhang, 2020 [187])	Deep representation learning for infection/disease prediction using microbiome data
mAML (F. Yang & Zou, 2020 [188])	Automated human disease classification through reproducible models
DeepARG (Arango-Argoty et al., 2018 [189])	Utilizes deep learning to predict novel antibiotic resistance genes
PaPrBaG (Deneke et al., 2017 [190])	Pathogenicity prediction, reliable even at low genomic coverage

## Data Availability

Not applicable.

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
