# Peer review of "Computational Metagenomics: State of the Art"

_ijms, 2025, doi:10.3390/ijms26189206_

Round 1

Reviewer 1 Report

Comments and Suggestions for Authors

Review For IJMS

The review by authors are a comprehensive review on the topic of human microbiome analysis, including the library preparation to the downstream analysis. In general, I think the manuscript is well-prepared and the information in this review is important to accelerate the microbiome analysis in the scientific community.

The following are some suggestions for improving the manuscript for the authors’ consideration.

  • Overview of Metagenomics and Microbiome Analysis

Line 100: Here, the authors mark genetic material as ‘DNA’, however, the author consider virus in the following context. Is RNA virus can be sequenced in the metagenomic analysis? Please clarify.

Line 104: the number of reference is not in sequence. ‘[1,2]’ could be more appropriate.

Section ‘Gene Prediction and Annotation’

Line 433 The author introduce Prodigal which is used for prokaryote gene prediction, I suggest authors also introduce some software which can used for eukaryote gene prediction, as fungi could be a important component of microbiome. Comparison of gene prediction software could also help readers to design their research.

L436-L437 please delete blank between DIAMOND and the following words.

The Authors, in this section make comprehensive comparison on function annotation tools. Could the authors here discussed the common limitation on these tools, and the potential impact on the downstream analysis? What is the most important direction that refine the function annotation ?

3.2.1. Differential abundance

How to analyze differential abundance need to further describe here, as here the authors only present the normalization methods. The core idea/rationale to compare the abundance between samples need to present here.

3.4. Network and Interaction Analysis

The authors discussed microbial interaction inference here. I think this topic is useful. However, the authors focuses on the methods selection and make little discussion on the downstream analysis approaches/metrics. Add some brief introduction on the downstream analysis such as network topological analysis, the estimation on the stability/robustness of co-occurrence network could be useful.

The author reviewed the medical finding/advances driven by microbiome analysis. However, I think the metagenomic approaches described in this review need to be stressed in this part to show how metagenomic analysis lead the new findings.

Author Response

The review by authors is a comprehensive review on the topic of human microbiome analysis, including the library preparation to the downstream analysis. In general, I think the manuscript is well-prepared and the information in this review is important to accelerate microbiome analysis in the scientific community.

Response: 

We sincerely thank the reviewer for the thorough assessment of our manuscript and the constructive feedback provided. We carefully addressed all the points raised, including citation style, figure referencing and labeling, formatting issues, redundancy, and missing references. These aspects have now been revised and corrected in the updated version of the manuscript, which we believe has significantly improved in clarity and quality thanks to the reviewer’s insightful comments. To ease reading our responses appear in bold print.

The following are some suggestions for improving the manuscript for the authors’ consideration.

  • Overview of Metagenomics and Microbiome Analysis
  1. Line 100: Here, the authors mark genetic material as ‘DNA’, however, the authors consider viruses in the following context. Is RNA virus sequenced in the metagenomic analysis? Please clarify.

Response: 

We appreciate the reviewer’s observation. In the revised version, we have clarified that while metagenomics is often defined as the study of the collective DNA recovered directly from environmental samples, RNA viruses can also be analyzed.

Page 3 - Paragraph 4 - Lines 98 - 105

Consider exploring the hidden world of microorganisms that profoundly influence our health and environment. Metagenomics is the study of the collective genetic material of microorganisms recovered directly from environmental samples, typically referring to DNA. However, it is important to note that RNA viruses can also be studied within metagenomic frameworks by converting their RNA into complementary DNA (cDNA) prior to sequencing. This approach broadens the scope of metagenomics, enabling the characterization of both DNA-based and RNA-based microbial communities and providing a more comprehensive view of their diversity and ecological roles.

Line 104: the number of references is not in sequence. ‘[1,2]’ could be more appropriate.

Section ‘Gene Prediction and Annotation’

  1. Line 433 The authors introduce Prodigal which is used for prokaryote gene prediction. I suggest authors also introduce some software which can be used for eukaryote gene prediction, as fungi could be an important component of microbiome. Comparison of gene prediction software could also help readers to design their research.

Response: 

We thank the reviewer for this valuable suggestion. We agree that highlighting gene prediction tools for eukaryotes is relevant as fungi and protists represent a relevant fraction of microbiomes. In the revised version we added tools tailored for eukaryotic gene prediction in metagenomic contexts.

Page 14 - Paragraph 4 - Lines 485 - 499

Tools like Prodigal [55] are widely used to identify genes in assembled contigs, being highly optimized for bacterial and viral genomes, and thus remain the preferred choice in prokaryote-dominated microbiomes. Prodigal has consistently shown high accuracy in benchmark datasets, achieving >90% sensitivity and specificity in bacterial genomes and CAMI community challenges [48]. However, when eukaryotic microorganisms such as fungi constitute a significant component of the community, specialized tools are more suitable. For example, MetaEuk [56], benchmarked on large-scale marine microbiomes, demonstrated high sensitivity (~92%) and precision (>85%) for gene recovery from fragmented assemblies by leveraging homology-based searches. Similarly, EukMetaSanity  [57], tested on simulated fungal metagenomes and environmental datasets, integrates multiple predictors and achieved robust gene annotation with F1-scores of ~0.85–0.90. Therefore, the choice of software should be guided by both the taxonomic composition of the dataset and its complexity: Prodigal for prokaryotes and viruses, and MetaEuk/EukMetaSanity when eukaryotic members such as fungi or protists are of interest.

L436-L437 please delete the blank space between DIAMOND and the following words.

  1. The Authors, in this section make comprehensive comparison on function annotation tools. Could the authors here discuss the common limitation on these tools, and the potential impact on the downstream analysis? What is the most important direction that refine the function annotation?

Response:

We appreciate this pertinent observation. As you mentioned, functional analysis methods have particular assumptions and limitations that can impact downstream analysis and therefore, their interpretation. The following text focuses on outlining the limitations and underlying assumptions of these analyses. 

Page 16 - Paragraph 4 - Lines 576 - 621

Although functional analysis serves as an inference of the possible metabolic capabilities of the studied microbiome, they can help in the hypothesis generation in newly studied populations. However, when describing/generating this hypothesis a few pointed assumptions and limitations must be considered. 

Firstly, short read metagenomic approaches assume that these short sequences contain sufficient information to be mapped to its origin gene or an equivalent ortholog and, therefore, a correctly associated function. Secondly, these approaches assume that sequences obtained from intergenic regions or from orthology groups with unknown function will not be erroneously mapped [77].  Given that up to date most metagenomic studies approaches still rely on these approaches for large scale studies due to their cost-effectiveness and high accuracy [78,79], these assumptions must be carefully considered, as they propagate to downstream analysis [77]. 

That said, modern metagenomic workflows such as DRAM and HUMAnN3 as part of bioBakery 3 used to process short read sequences, prioritize assembly and the reconstruction of MAGs before carrying out functional annotation and profiling. This strategy markedly reduces annotation ambiguity, since contigs and MAGs provide a broader genomic context than individual reads, thereby improving functional assignment [36]. In addition, the use of long reads has enabled the capture of full length gene sequences and even entire operons owing to the generation of longer contigs during assembly. This has significantly enhanced the recovery of near complete and high quality MAGs, which are more informative for downstream functional analysis. Nevertheless, the elevated cost and infrastructure demands of long read sequencing can limit its widespread adoption, particularly in large scale or clinical studies. To address this limitation, hybrid approaches  such as hybridSPADES [80] , OPERA-MS [81] and MetaPlatanus [82] have been increasingly recommended, as they combine the cost effectiveness and depth of short reads with the assembly advantages of long reads, facilitating the recovery of longer contigs and more complete MAGs but also improve the overall mapping rates to bacterial genomes, providing a more reliable foundation for functional characterization [10,79].

It is important to highlight that, even high quality MAGs are subject to errors that can arise from DNA extraction all the way to the assembly processes, leading to the loss of sets of genes and the overall representation of them in a particular microbiome [78]. Furthermore, although functional databases have significantly expanded in recent years, a substantial proportion of genes in microbiomes remain annotated as “hypothetical” [83]. This implies that functional inference is still largely dependent on homology to orthologous genes from reference organisms, many of which are biased towards well-studied eukaryotes or a limited set of prokaryotic models. Consequently, while functional profiling is a powerful tool for exploring metabolic potential, it is essential to acknowledge its inferential nature, the risks of over-interpretation, and the propagation of annotation errors in complex or under-characterized microbiomes. 

Moving forward, the most important direction for refining functional annotation lies in the integration of high quality MAGs with long read and hybrid assemblies, coupled with the continuous expansion and curation of functional databases to reduce the proportion of genes classified as hypothetical. In parallel, the incorporation of complementary omics approaches such as metatranscriptomics, metaproteomics, and metabolomics will be critical to validate gene functions beyond sequence homology, thus bridging the gap between inferred metabolic potential and experimentally supported functional activity.

  1. 3.2.1. Differential abundance

How to analyze differential abundance needs to be further described here, as here the authors only present the normalization methods. The core idea/rationale to compare the abundance between samples need to be presented here.

Response:

We agree with your observation. In the revised version of this section, we have added a brief rationale for the use of differential abundance analysis, incorporated a citation to the benchmarking study by Nearing et al that compared several differential abundance tools and included a recommendation to employ multiple methods in parallel to enhance the robustness of the results. 

Page 23 - Paragraph 1 - Lines 857- 920

The simultaneous presence of high dimensionality,  sparsity, over dispersion and compositionality poses significant challenges for downstream statistical analysis, requiring specialized frameworks to ensure robust and biologically meaningful description and inference. Therefore, appropriate data normalization constitutes a crucial step to account for the characteristics of microbiome data and improve the comparability of samples across data.

There are several strategies employed in normalization methods, some are based on data scaling such as Total Sum Scaling (TSS), Cumulative Sum Scaling (CSS), Trimmed Mean of M-Values (TMM) and Relative Log Expression (RLE) which adjust samples count data by a normalization factor to correct the different sequencing depth or global composition, and others like Centered-Log Ratio (CLR) and Variation Stabilizing Transformation (VST) transform relative abundances to a logarithmic space, eliminating the constant sum restriction. 

After normalization, differential abundance analysis aims to test whether microbial features differ significantly and the magnitude of change between clinical in their abundance across phenotypes or experimental groups   (Figure 3D).  To this end, several R packages like DESeq2 [141], LEfSe [142], edgeR [143], ALDEx2 [144] and ANCOM-BC2 [145], are commonly used. 

For instance, DESeq2 and edgeR, originally developed for differential gene expression in RNA‑Seq, are now frequently applied in microbiome studies. These tools leverage RLE and TMM normalization, respectively, and model counts using the Negative Binomial distribution to accommodate compositional biases. They offer high sensitivity in detecting differential taxa but struggle to control False Discovery Rate (FDR) in sparse microbiome datasets.

LefSe, on the other hand, normalizes count data using TSS or Counts Per Million (CPM), applies non parametric test and finally uses Linear Discriminant Analysis to estimate effect size, however this method is prone to false positives in high dimensional or sparse datasets.

In contrast, methods explicitly designed for compositional data, such as ALDEx2, which employs a Dirichlet multinomial model coupled with Monte Carlo sampling and performs a CLR transformation data and ANCOM-BC2, which uses a log transformed regression based approach and adds a regularization step to stabilize variance estimates, enhance the robustness of differential abundance analysis as they reduce false positives while maintain FDR control by accounting for technical and biological variation in compositional data.

Nearing et al. [146] systematically evaluated 14 differential abundance methods across several 16S rRNA gene datasets, revealing that count-based approaches such as edgeR and limma-voom tend to identify a much larger number of significant taxa compared to more conservative methods. In contrast, ALDEx2 and ANCOM produced fewer significant results, but the taxa they detected were far more reproducible across datasets, suggesting a lower false positive rate. Furthermore, this author evaluated the discriminatory potential of individual ASVs  identified by each tool through ROC curves and the Area Under the Curve (AUC). They observed that ASVs identified by ALDEx2 and ANCOM-II exhibited the highest mean AUROC across datasets, both when using relative abundances and CLR-transformed data, although these methods occasionally failed to detect any significant ASVs even in cases where other tools achieved high AUROCs (e.g., 0.8–0.9). To further evaluate performance, the authors compared precision, recall, and F1-scores of different methods using AUROC thresholds. At a threshold of 0.7, ALDEx2 and ANCOM-II achieved the highest precision values (often approaching 1.0), but this came at the cost of very low recall, especially when contrasted with tools such as LEfSe and edgeR, which displayed much higher recall but lower precision. When CLR data were used, limma-voom and the Wilcoxon test applied to CLR abundances showed among the highest F1-scores, reflecting a better balance between sensitivity and specificity. At a more stringent AUROC threshold of 0.9, most tools showed relatively high recall, but precision declined sharply across all frameworks, consistent with the expectation that few taxa exceed such strong discriminatory performance.

The trade-offs between precision, recall, and F1-score, as highlighted by Nearing et al. [146], emphasize the need for careful tool selection, prevalence filtering, and consideration of both statistical robustness and biological interpretability when drawing conclusions from microbiome differential abundance analyses. More generally, it is advisable to employ multiple methods in parallel and to focus on features consistently identified across tools, while keeping in mind the specific assumptions and limitations inherent to each framework.

  1.  3.4. Network and Interaction Analysis

The authors discussed microbial interaction inference here. I think this topic is useful. However, the authors focus on the methods selection and make little discussion on the downstream analysis approaches/metrics. Add some brief introduction on the downstream analysis such as network topological analysis, the estimation on the stability/robustness of co-occurrence network could be useful.

Response:

We thank the reviewer for this valuable observation. We agree that our initial draft overemphasized the selection of inference methods while offering limited perspective on downstream analyses. In the revised version, we now provide a brief introduction to the types of analyses that can be conducted once networks are inferred and their implication on the microbiome studied either for co-occurrence or metabolic interaction networks.

Page 27 - Paragraph 1 - Lines 1076- 1094

Importantly, downstream analyses of these inferred networks provide an additional layer of insight. Network theory offers a suite of topological metrics such as degree, betweenness, modularity, assortativity or nestedness, that help identify keystone taxa that interact with several other taxons or functions as well as the modular organization of communities [174,178]. These properties are linked to ecological and functional implications: for example, higher modularity often confers resilience to the microbial community by containing disturbances within only few taxa, while betweenness highlights taxa that mediate information or metabolite flow across modules, thereby influencing nutrient exchange and cross-feeding relationships. Similarly, assortativity captures whether highly connected taxa preferentially interact with other highly connected taxa,a  pattern that can influence robustness by reinforcing clusters or by increasing vulnerability to targeted perturbations in its absence [179]. Beyond structural attributes, these metrics also illuminate functional consequences: for instance, modules can correspond to metabolic guilds involved in short-chain fatty acid production, nitrogen cycling or bile acid metabolism which disruption may explain shifts in host physiology or disease progression.  Recent reviews emphasize that these stability-oriented metrics provide a mechanistic bridge between network architecture and emergent properties such as resilience, productivity, and metabolic complementarity, and are increasingly used to evaluate how microbial communities respond to environmental stress or host-associated disturbances [179,180].

  1. The authors reviewed the medical findings/advances driven by microbiome analysis. However, I think the metagenomic approaches described in this review need to be stressed in this part to show how metagenomic analysis leads to the new findings.

Response:

We agree that the "Applications" section would be significantly strengthened by more explicitly connecting the medical findings to the metagenomic approaches that enabled them. As suggested, we have revised this section to better highlight how these analytical methods have driven the discoveries discussed.

Page 33 - Paragraph 1 - Lines 1299- 1582

5. Applications in Human and Environmental Health

The human microbiome is composed of diverse microbial communities, including bacteria, archaea, viruses, phages, and fungi, which resides in different body sites such as the oral cavity, skin surface, intestinal tract, oesophagus, lungs, and more [196]. Microbial colonization is not uniform across the body; each body compartment contains its own microbiota, and even within the same site, microbial composition can vary depending on the specific area of sampling [197].

Microbes interact with one another both within the same species and across different species, genera, families, and even domains of life. These symbiotic relationships influence microbial fitness, population dynamics, and functional capacities. Interactions can be beneficial (mutualism, synergism, or commensalism), detrimental (amensalism, including predation, parasitism, antagonism, or competition), or neutral, where no significant effect on microbial function is observed [197].

Microbial colonization begins at birth and is influenced by factors such as the mode of delivery [198,199] and feeding practices [200]. The microbiome is dynamic and shaped by various elements, including aging [201], diet [202], antibiotic exposure [203,204], disease [205,206], and other environmental influences.

Beyond microbial interactions, the microbiome plays a crucial role in maintaining host health by regulating metabolism, immunity, and homeostasis. Several key functions have been attributed to the core microbiome, including polysaccharide digestion, immune system development, protection against infections, vitamin synthesis, fat storage, angiogenesis regulation, and even behavioral development [207].

However, when dysbiosis occurs—characterized by shifts in microbial composition, reduced diversity, and functional imbalances—these once-beneficial interactions can contribute to the onset of various diseases. Metagenomic approaches have been indispensable in moving beyond simple associations to dissecting these complex interactions.

5.1 Microbiome and Disease

Computational metagenomics and microbiome analysis have been applied in a variety of studies to explore the relationship between microbial communities and disease development. Early studies, primarily employing targeted 16S rRNA gene sequencing, were instrumental in characterizing broad shifts in the taxonomic composition of microbial communities in various tissues and fluids. While foundational, this approach is limited to taxonomic profiling and often cannot resolve below the genus level. The advent of shotgun metagenomics provided a much deeper view by enabling not only higher-resolution taxonomic classification but also functional profiling of the entire gene content of a community and the reconstruction of MAGs. More recently, the emergence of LRS is further revolutionizing the field by allowing for the assembly of complete, closed genomes directly from complex samples, which is critical for strain-level resolution and understanding the genomic context of functional genes [28,208].

Metabolic diseases

The gut microbiota has been suggested to play an important role in obesity, potentially influencing energy absorption, central appetite regulation, fat storage, chronic inflammation, and circadian rhythms [209]. Although the Firmicutes/Bacteroidetes ratio was initially proposed as a hallmark of obesity [210,211], subsequent meta-analyses and large-scale studies failed to replicate this association consistently.  The inconsistency of this finding across populations highlighted the limitations of low-resolution taxonomic markers. The transition to shotgun metagenomics shifted the focus from broad phylum-level changes to the functional capacity of the microbiome. These studies revealed that obesity is more consistently associated with a reduction in overall microbial gene richness and functional diversity, particularly in pathways related to butyrate production and carbohydrate metabolism. By reconstructing MAGs from obese and lean individuals, researchers have been able to link specific microbial lineages, such as members of the Christensenellaceae family, to lean phenotypes [212–215].

While these short-read shotgun studies provide powerful functional insights, they often yield fragmented MAGs. The use of long-read sequencing is now helping to overcome this, making it possible to assemble more complete genomes of key metabolic players like Akkermansia muciniphila, which allows for strain-level analysis of its role in metabolic health [28]. This progression demonstrates how advancements in metagenomic methods have enabled a more nuanced understanding of the microbiome's role in metabolic disease, moving from simple taxonomic ratios to detailed functional and strain-level insights.

Liver diseases

Some evidence suggests a strong association between gut dysbiosis and the onset and progression of metabolic dysfunction-associated steatotic liver disease (MASLD) [216].  Shotgun metagenomic studies have been crucial in identifying these links. For instance, by comparing the functional gene content of patients with MASLD to healthy controls, researchers have identified an overabundance of pathways related to inflammation and lipogenesis, often linked to genera like Escherichia and Streptococcus. The ability to perform whole-genome assembly from metagenomic data has also allowed for the reconstruction of MAGs of these pathogenic bacteria, revealing virulence factors that may contribute to liver inflammation. Research by Solé et al. used quantitative shotgun metagenomics to demonstrate a clear reduction in gene richness as cirrhosis progresses, directly linking the functional capacity of the microbiome to the severity of liver dysfunction. The next frontier in this area is using long-read sequencing to assemble complete viral genomes (the virome), as bacteriophages can modulate bacterial populations and may play a key role in the progression of liver disease [208,217].

Pregnancy and Reproductive Disorders

Recent research describes the role of the gut microbiome in pregnancy, particularly its influence on maternal and fetal health outcomes. Early investigations using 16S rRNA sequencing established correlations between broad shifts in the vaginal microbiota, such as a decrease in Lactobacillus dominance, and adverse pregnancy outcomes. However, shotgun metagenomics has been critical in moving beyond taxonomic associations to understanding functional potential [218]. By analyzing the collective gene content, researchers have been able to link changes in metabolic pathways—such as amino acid and carbohydrate metabolism—to conditions like preeclampsia and gestational diabetes mellitus [208]. 

Furthermore, studies [219,220] have shown that dysbiosis of gut microbiota in women could be associated with Polycystic Ovary Syndrome (PCOS).  Shotgun metagenomics has revealed not just a change in species, but a shift in functional capacity toward inflammatory pathways. A key challenge in this area is the high degree of strain-level diversity among critical commensals like Lactobacillus. Different strains of the same species can have vastly different probiotic or pathogenic potentials. Therefore, modern approaches using long-read sequencing (LRS) to assemble complete genomes and achieve true strain-level resolution are becoming essential for determining which specific lineages are protective versus detrimental during pregnancy [8]. This highlights a critical application where moving from species-level profiling to high-resolution strain identification is necessary to translate research findings into clinical diagnostics or therapies.

Neurological disorders

The microbiome also plays a significant role in neurodevelopmental disorders and behavior. The gut-brain axis is one of the most exciting frontiers in microbiome research, and metagenomics provides the primary toolkit for its exploration. Alterations in the gut microbiota have been observed in individuals with autism spectrum disorder (ASD) and other neurodevelopmental disorders. Again, initial findings from 16S rRNA profiling identified taxonomic markers, such as an increased abundance of Clostridium species in some ASD cohorts. However, the mechanism remained elusive [221–223]. 

The adoption of shotgun metagenomics provided the crucial next step by linking these taxonomic shifts to functional changes. By analyzing the microbial gene content, studies have identified depletions in pathways for synthesizing key neurotransmitters (for example, GABA and serotonin precursors) and short-chain fatty acids (SCFAs) like butyrate, which are known to have neuroprotective effects. To understand if these functional deficits are due to the absence of genes or simply a lack of their expression, researchers are increasingly turning to metatranscriptomics. This approach directly measures microbial gene expression (RNA), revealing which metabolic pathways are actively transcribed by the gut community. This allows a distinction between the genetic potential (DNA) and the real-time functional activity (RNA), which is critical for understanding dynamic conditions like ASD [224]. A comprehensive review by Liu et al. (2024) summarizes how these multi-omic metagenomic approaches are building a body of evidence linking the gut microbiome to a wide range of neurodegenerative diseases [225].

Inflammatory Skin Disorders

In dermatology, recent studies have linked dysbiosis of the cutaneous and intestinal microbiomes with skin-associated diseases, such as psoriasis [226], atopic dermatitis [227], acne vulgaris, rosacea, alopecia areata, and hidradenitis suppurativa [228]. Specifically, a systematic review by Widhiati et al. [229] explored the associations between the gut microbiome with inflammatory skin disorders, suggesting that Bifidobacterium plays an essential role as anti-inflammation bacteria, and Proteobacteria and Enterobacteria impact inflammation in inflammatory skin disorders.

The reconstruction of these MAGs has been vital for identifying strain-specific virulence factors and metabolic functions that correlate with disease states like acne or atopic dermatitis. Moreover, metatranscriptomics is being used to determine which microbial pathways are active on the skin during inflammatory flares versus periods of health. By sequencing the expressed RNA, it is possible to identify if, for example, certain strains of S. aureus are actively transcribing toxins or if commensal microbes are expressing anti-inflammatory molecules [224]. This functional insight provides a much more dynamic picture of the skin microbiome's role in disease than what can be gleaned from DNA-level surveys alone.

Autoimmune Diseases

Inflammatory bowel disease (IBD), a chronic relapsing-remitting disorder of the gastrointestinal tract, is closely linked to intestinal microbiota. IBD represents a paradigm for the power of advanced metagenomic analysis. The field has progressed significantly from early 16S rRNA studies, which first described a general state of dysbiosis in IBD, characterized by a loss of butyrate-producing bacteria like Faecalibacterium prausnitzii. While important, these studies could not explain the functional consequences of this dysbiosis [230].

The breakthrough came with the application of integrated, multi-omic approaches, exemplified by large-scale longitudinal studies like the Integrative HMP [231]. By combining shotgun metagenomics (MGX), metatranscriptomics (MTX), metaproteomics (MPX), and metabolomics, this study provided a holistic view of the IBD gut ecosystem.

  • Metagenomics confirmed the depletion of key commensals and provided the genomic blueprints (via MAGs) of the community members.
  • Metatranscriptomics revealed that during disease flares, the remaining microbes, including pathobionts like Ruminococcus gnavus, were transcriptionally hyperactive, indicating a highly stressed and inflammatory environment [224,231].
  • Metaproteomics complemented this by providing direct evidence of which proteins—the functional workhorses of the cell—were being produced, confirming that inflammatory and stress-response pathways were highly active [232].
  • Metabolomics measured the downstream biochemical output, identifying clear shifts in bile acid metabolism and SCFA production that correlated with the functional changes observed at the gene and protein level.

This integrated metagenomic approach revealed that IBD is not just a change in who is there, but a fundamental rewiring of the entire molecular activity of the microbiome. It is the ability to combine these different layers of metagenomic data that is leading to new diagnostic biomarkers and therapeutic targets for IBD.

5.2 Therapeutic Applications

The clinical application of microbiome-targeted therapies relies heavily on metagenomic tools for both development and monitoring.

Probiotics

Probiotics are live microorganisms that, when administered in adequate amounts, confer a health benefit on the host [233]. The effectiveness of probiotics is highly strain-specific. Therefore, simply using 16S rRNA sequencing to confirm the presence of a Lactobacillus species is insufficient. Shotgun metagenomics, and particularly long-read sequencing, is required for the high-resolution tracking of specific probiotic strains to confirm their engraftment and functional activity in the gut [8].

Some of the species from which strains with probiotic characteristics have been isolated are Lacticaseibacillus casei, Lacticaseibacillus paracasei, Lacticaseibacillus rhamnosus, Lactiplantibacillus plantarum, Lactobacillus acidophilus, Levilactobacillus brevis, Ligilactobacillus salivarius [234].

These are commonly found in probiotic formulations, such as dietary supplements or medicine, and in foods like fermented dairy products or other fermented foods [234].

According to Wang et al. [235], probiotics play a significant role in maintaining and modulating gut microbiota composition. They can adjust the structure of human intestinal microorganisms and inhibit the colonization of pathogenic bacteria in the intestine. This inhibition occurs through various mechanisms, including stimulation of epithelial barrier function, production of antimicrobial substances, restriction of pathogenic access to nutrient resources, and competition for binding sites. Additionally, probiotics secrete organic acids—such as butyric acid, acetic acid, and propionic acid—during carbohydrate fermentation. These acids lower the intestinal pH, thereby creating an environment that inhibits the growth of harmful bacteria.  

Probiotics also contribute to intestinal health by enhancing the protective layer of intestinal mucosa, thereby strengthening the intestinal barrier and improving immune function. Notably, butyric acid, a key metabolite produced by probiotics, promotes oxygen consumption in the intestinal epithelium. This process increases the expression of barrier-protective hypoxia-inducible factor (HIF) target genes and helps maintain HIF stability, which is essential for intestinal homeostasis [235].

Furthermore, probiotics enhance immune responses by increasing the number and functionality of macrophages and dendritic cells (DCs) in the lamina propria. They also promote the maturation of humoral immunity by stimulating the production of immunoglobulin A (IgA) antibodies, which play a critical role in mucosal defense [235]. 

Given their ability to modulate gut microbiota, probiotics have emerged as a promising therapeutic approach for managing various microbiota-related diseases. Evidence suggests that certain probiotic strains are effective in preventing antibiotic-associated diarrhea in both adult and pediatric populations. Acute gastroenteritis remains the original and probably the most well-established clinical indication for probiotic use [236]. However, there are still issues with dose and timing that require further research. 

Prebiotics

A prebiotic is a substrate that is selectively utilized by host microorganisms, conferring a health benefit [237]. Among the most studied prebiotics are oligosaccharide carbohydrates, of which mainly include xylooligosaccharides (XOS), galacto-oligosaccharides (GOS), lactulose and inulin and its derived fructose-oligosaccharides [238]. These compounds are selectively fermented in the colon, leading to the production of short chain fatty acids (SCFAs) and biomass, decreasing the pH value of the intestinal environment. Therefore, they create favorable conditions for the growth and activity of beneficial bacteria, such as bifidobacteria and lactobacilli (e.g., L. plantarum, L. paracasei, B. bifidum), while simultaneously inhibiting the proliferation of pathogens (Clostridium perfringens., Escherichia Coli, Campylobacter jejuni, Enterobacterium spec., Salmonella enteritidis or Salmonella typhimurium), thereby contributing to improved human health [239].

Prebiotics occur naturally in a variety of foods, including wheat, onions, bananas, honey, garlic, and leeks [234]. Their health benefits include improving the regulation of immunity, resisting pathogens, influencing metabolism, increasing mineral absorption, and enhancing health [238].

Faecal microbiota transplantation

Faecal microbiota transplantation (FMT) is a treatment that involves the administration of a minimally manipulated microbial community from the stool of a healthy donor into a patient’s intestinal tract to restore normal gut microbiota function [240]. 

FMT is an important therapeutic option for Clostridioides difficile infection (CDI). Its use in treating recurrent CDI has been shown to be well tolerated and effective when primary treatment has failed [241].

Metagenomics is the primary tool for assessing the success of FMT. Shotgun sequencing is used to perform deep taxonomic and functional profiling of both the donor's and recipient's microbiota. Post-transplantation, this analysis is repeated to track the engraftment of donor strains and the restoration of key metabolic pathways that were deficient in the recipient. This provides a quantitative, data-driven measure of therapeutic success [242].

Precision medicine

As mentioned previously, the function, composition, and growth dynamics of the gut microbiome are associated with many host physiological and pathological states. This evidence highlights the potential of the gut microbiome in precision medicine and individualized treatment. 

Non-invasive sampling methods and decreasing profiling costs make it a feasible tool for early diagnosis and disease risk assessment. Integrating features related to the composition of the gut microbiome with other known clinical risk factors may potentially enhance early disease detection [243].

This will be driven by the integration of multi-omics data, similar to what has been done for IBD [231]. By generating a comprehensive metagenomic, metatranscriptomic, and metabolomic profile of a patient, it may become possible to computationally predict their response to a specific diet, probiotic, or therapeutic drug. The development of robust, high-throughput computational pipelines for integrating these complex datasets is therefore a critical area of ongoing research that will be essential for bringing precision microbiome medicine into the clinic [208].

However, many challenges and limitations still need to be addressed for microbiome profiling to be fully integrated into common medical practice. Zmora et al. [243] state that the microbiome shows a remarkable degree of inter-personal variability, with even intra-personal fluctuations that tend to oscillate diurnally. Additionally, differences in sample collection and analysis techniques, reagents, and parameters may introduce variations into microbiome results, potentially adding biases to the interpretation of microbiome-based data.

The field must develop robust and standardized protocols for sample collection, sequencing, and analysis to improve the reproducibility of results and reduce biases.

Reviewer 2 Report

Comments and Suggestions for Authors

The article makes a very solid impression. You have laid out a beautifully structured foundation that will be extremely useful for researchers who are just beginning to immerse themselves in this complex area. Your deep understanding of the subject and ability to work with large amounts of information is clear.Having said that, I have a few suggestions that could strengthen the impact of your work, turning it from a very good review into a definitive one.Firstly, I would like to see a more critical examination of the methods being compared. The reader is not offered any guidance when faced with a choice in their own work. It would be extremely useful to base performance claims on specific metrics such as accuracy, sensitivity and false discovery rate, using data from large benchmarking studies such as CAMI and MBARC. For example, it is difficult to state that one tool is more accurate than another; however, it would be possible to indicate that, on a given dataset, one tool achieved a recall of 95% versus 80%, albeit at the cost of increasing the FDR by 10%. Finally, as the article is intended as a review of the current state of the art, the addition of a section dedicated to the most recent and promising trends that will determine the future of the field would be greatly beneficial. These areas include single-cell metagenomics, which bypasses assembly problems, real-time analysis of nanopore sequencer data for epidemiological surveillance and pangenome graphs for accounting for variability. A brief but comprehensive overview of these technologies would demonstrate that the review considers not only the present, but also the future.In addition, the review contains a large number of typos that need to be corrected.

To the specific lines:

line 57: The citation "(Lema et al., 2023)" should be formatted as "[2]";

line 241: Figure 2 is not referenced in the main text, unlike the other figures;

lines 243-244: It is not clear where points A) and B) are shown in Figure 2;

lines 246-254, 256-266, 268-289: Unnecessary italics. Please use regular font;

lines 325-329: The text describing the MIMAG standarts is almost completely repeated on lines 337-340;

lines 329 and 340: I did not find (Bowers et al., 2017) in the list of references;

line 409: Misprint "Kracken2" -> "Kraken2";

line 410: Remove the unnecessary period after "[44,45].";

line 413: You probably meant "achieve", not "archive". And also "resolution", not "resolutions";

line 427: Misprint "MetaPlAn" -> "MetaPhlAn";

line 436: Unexpected line break between "DIAMOND" and "[54]";

line 451: "...aiming to bridge the gap" would be grammatically more correct than "...in aims to bridge the gap";

line 467: Misprint "gen transfer" -> "gene transfer";

line 475: "which are capable of annotating" would be grammatically better than "...them being capable of annotate";

line 483: Misprint "BlasKOALA" -> "BlastKOALA";

line 509: Remove the duplicated period after "previous studies";

line 512: The "Dataset" column lists vastly different and incompatible data sets for many tools (differing in size and origin). This makes a direct comparison of "Memory use" and "Running time" across tools impossible;

line 541: SnakeWRAP is actually implemented in Snakemake, not Nextflow;

line 564: "biomedicine field" would be better than "biomedicine area";

line 712: "this datasets" -> "these datasets";

line 715: "this features" -> "these features";

line 717: Misprint "in wich" -> "in which";

line 732: "Variation Stabilization Transformation" -> "Variation Stabilizing Transformation";

line 754: "data as tested Nearing (2022)" -> "data, as tested by Nearing et al. (2022)";

line 916: "Receiver Operating Characteristic curve" -> "Receiver Operating Characteristic (ROC) curve";

line 933: "The field of machine learning is fairly vast" -> "The field of machine learning encompasses a wide range of techniques";

line 986: "Table 4" -> "Table 5" ("Table 4" already exists on page 14);

line 996: Unexpected line break between "existing databases" and "AI-based tools";

line 998: The claim that "AI tools can process... rapidly with smaller models, significantly reducing storage and memory usage" is an overgeneralization. Although certain models are efficient, many contemporary deep learning architectures, such as large language models and transformers, are famously resource-intensive and require vast amounts of memory. This statement needs to be softened or made more precise;

line 1346: "...the the Minimal Information" -> "...the Minimal Information";

Comments on the Quality of English Language

Esteemed colleagues,
The primary issue with the manuscript is the presence of numerous typos. It is imperative that a meticulous proofreading process is undertaken to rectify the identified technical deficiencies, thereby elevating the work's overall quality and professional standards.

Author Response

The article makes a very solid impression. You have laid out a beautifully structured foundation that will be extremely useful for researchers who are just beginning to immerse themselves in this complex area. Your deep understanding of the subject and ability to work with large amounts of information is clear.Having said that, I have a few suggestions that could strengthen the impact of your work, turning it from a very good review into a definitive one. 

Response: 

We sincerely thank the reviewer for the careful assessment of our manuscript and the constructive feedback provided. All the issues raised regarding citation style, figure referencing and labeling, formatting, redundancy, and missing references have been carefully revised and corrected in the updated version of the manuscript. To ease reading our responses appear in bold print.

Firstly, I would like to see a more critical examination of the methods being compared. The reader is not offered any guidance when faced with a choice in their own work. It would be extremely useful to base performance claims on specific metrics such as accuracy, sensitivity and false discovery rate, using data from large benchmarking studies such as CAMI and MBARC. For example, it is difficult to state that one tool is more accurate than another; however, it would be possible to indicate that, on a given dataset, one tool achieved a recall of 95% versus 80%, albeit at the cost of increasing the FDR by 10%.

Response: 

We thank the reviewer for this valuable comment. We agree that our initial comparison lacked the level of critical evaluation needed to guide readers in tool selection. In the revised version, we now incorporate performance metrics such as F1-score and False positives rates for tools that have been evaluated side by side in large benchmarking efforts based on CAMI datasets. In addition, we have included a forward-looking paragraph emphasizing the pressing need for comprehensive benchmark studies targeting functional annotation tools, which remain comparatively under-evaluated.

Page 3 - Paragraph 4 - Lines 442- 575

Whole Genome Metagenomics

In recent years, whole genome metagenomics has gained prominence due to improvements in accuracy of both short and long sequencing technologies as well as the rapid expansion and enrichment of reference databases. This has promoved the production and enhancement of numerous tools that allow taxonomic classification and profiling. Short reads oriented tools such as Bracken [37], Kraken 2 [31], Centrifuge [32] and CLARK/CLARK-S [38,39] perform k-mer based matching against reference databases. This strategy enables high-throughput processing of large datasets (< 1 hour) if high memory computational resources are provided ( > 100 GB of memory). 

Across tools, Kraken 2 has proven to offer an effective balance between speed and accuracy. On the CAMI2 gastrointestinal dataset [40] it achieved a precision of 0.94 coupled with fewer than 1% false positive and an impressive F1-score of 0.97, highlighting its robustness for species level assignments. In parallel, Centrifuge reached a very similar precision of 0.93, however it exhibited a rate of false positives around the 3% rate, while still obtaining a 0.97 F1-score [41]. From a practical standpoint, Kraken 2 may be favored in scenarios where minimizing false positives is critical, such as clinical diagnostics or studies requiring stringent taxonomic confidence. In contrast, Centrifuge offers advantages in settings where maximizing recall and broad detection are priorities, for instance in exploratory surveys of highly diverse or poorly characterized communities, even at the cost of a modest increase in false positives.

Although these tools can achieve species level resolution, their k-mer based approach often leads to substantial generation of false positives when the sample contains closely related taxa or organisms present at very low abundance (~0.05-1%). To mitigate this, researchers often apply abundance thresholds, focusing on retaining only taxa with relative abundances > 1% depending on study objectives.

When computational resources are limited, marker-based approaches such as MetaPhlAn  [42] mOTUs [43], are recommended. These tools rely on databases comprising a large number of gene families rather than complete reference genomes;however, the absence of full genome context can result in a high proportion of false positives. For this reason, the use of the most recent versions of these tools is advised. mOTUs 3  [44] incorporates 5,165 metagenomes from human associated microenvironments and substantially improves species level precision and recall, obtaining a F1-score ranging from 0.78 to 0.85 across several CAMI2 human microbiome datasets [40]. MetaPhlAn 4 [45] expands its database to 19.5 metagenomes from diverse body sites, which has allowed it to obtain F1-scores of 0.80 to 0.85 when using the OPAL framework [46] and a nearly perfect 0.95 score when employing genomes from the SGB organization as reference when this method is applied in several CAMI2 human microbiome datasets  [40]. Furthermore, the latest release (v4.2.2) of MetaPhlAn 4 adds support for long read sequence profiling through Minimap2-based alignment.

2.2.4.2 Gene Prediction and Annotation

Following taxonomic classification, genes are predicted and functionally annotated to understand microbial activity within communities. Annotation involves assigning function to gene sequences based on reference databases:

Gene Prediction

Tools like Prodigal [47] are widely used to identify genes in assembled contigs, being highly optimized for bacterial and viral genomes, and thus remain the preferred choice in prokaryote-dominated microbiomes. Prodigal has consistently shown high accuracy in benchmark datasets, achieving >90% sensitivity and specificity in bacterial genomes and CAMI community challenges [40]. However, when eukaryotic microorganisms such as fungi constitute a significant component of the community, specialized tools are more suitable. For example, MetaEuk [48], benchmarked on large-scale marine microbiomes, demonstrated high sensitivity (~92%) and precision (>85%) for gene recovery from fragmented assemblies by leveraging homology-based searches. Similarly, EukMetaSanity  [49], tested on simulated fungal metagenomes and environmental datasets, integrates multiple predictors and achieved robust gene annotation with F1-scores of ~0.85–0.90. Therefore, the choice of software should be guided by both the taxonomic composition of the dataset and its complexity: Prodigal for prokaryotes and viruses, and MetaEuk/EukMetaSanity when eukaryotic members such as fungi or protists are of interest.

2.2.4.3 Functional annotation

Functional annotation, as described by Wang [50], is a computational process that assigns putative functions to predicted genes discovered during gene discovery. Shotgun metagenomics enables the reconstruction of contigs and bins, facilitating detailed functional analysis of microbial communities. In contrast, marker gene approaches primarily provide taxonomic information and can only infer potential functions due to limited genomic data. Therefore, while marker gene methods offer insights into community composition, shotgun metagenomics is more effective for accurately characterizing the functional potential of microbiomes.

Marker gene approaches primarily offer taxonomic insights  but are limited in their ability to determine functional capabilities due to the scarcity of genomic data. To address this limitation, various computational tools have been developed to leverage marker gene information aiming to bridge the gap between taxonomic profiling and functional characterization by predicting metabolic pathways and biological processes based on marker gene distributions. In this regard, tools like PICRUSt2 [51], Tax4Fun2 [52], BugBase [53] and PanFP [54] have been developed. When evaluating their accuracy, however, clear differences emerge. In simulated datasets, PICRUSt2 demonstrated one of the lowest false positive rates (FPR), averaging 2.94%, while MicFunPred also performed conservatively with an average FPR of 6.86%. In contrast, Tax4Fun2 exhibited the highest level of false predictions, with an average FPR of 35.14%, highlighting its tendency to overestimate functional content [55]. These results emphasize that while marker gene-based approaches can provide functional predictions, their reliability varies considerably across tools and should be interpreted with caution. 

It’s important to remark that these tools are inherently limited by their reliance on 16S gene profiles and reference genomes. These methods extrapolate potential gene content based on phylogenetic similarity, as is the case of PICRUSt2 and Tax4Fun2 or pangenome reconstructions as PanFP, assuming that closely related taxa necessarily share similar functional capabilities. However, this approach overlooks intra-species variability, horizontal gene transfer events and strain specific adaptations. As a result, the inferred profiles represent approximations rather than measures of genetic content.

In contrast, whole genome sequences are assembled, to be further functionally annotated by comparing coding sequences with databases containing information on genes, proteins, and metabolic pathways [56]. This approach thus enables a comprehensive evaluation of the functional characteristics of microbial communities as they analyse the actual content of genes, and pathways present in a sample [57–59].

In this regard, tools like Prokka [60] and InterProScan [61] have been tested to annotate test genomes, which are capable of annotating between 75 and 90% of the individual genomes in less than 3 minutes. However, the reason for this performance is due to the limited database they use as reference, which can be an inconvenience when the samples analyzed contain multiple metagenomes. In contrast, tools like MicrobeAnnotator [62], which has higher computational demands, is able to annotate ~92% of individual genomes and can be used in complex samples as it combines multiple databases, offering an integral functional annotation of metagenomes.

Some tools like BlastKOALA and GhostKoala [63] are built as web services which offer different insights into genomic and metagenomic data. BlastKOALA delivers accurate annotations at the genome level, but its computational cost is substantial, requiring up to 80 minutes per genome. In contrast, GhostKOALA offers a much faster alternative, completing the annotation of full genome in less than 7 minutes.This gain in speed, however, comes at the expense of sensitivity, as GhostKOALA may miss novel functions or provide less comprehensive coverage compared to its counterpart.

Finally, alignment-based engines such as DIAMOND [64] and MMseqs2 [65] serve as the backbone for many pipelines. While DIAMOND delivers BLAST-like sensitivity at higher speeds, it loses sensitivity when dealing with highly divergent sequences or poorly conserved genes. MMseqs2 is even faster in large-scale comparisons, offering scalable annotation of millions of sequences with reduced memory footprints, yet this gain in efficiency can come at the cost of sensitivity, requiring careful parameter tuning and greater technical expertise to achieve optimal results.

Looking ahead, future work should prioritize systematic benchmarking of functional annotation tools such as InterProScan, Prokka, and Diamond. While taxonomic profiling has benefited from community-driven efforts like CAMI, functional annotation still lacks standardized large-scale evaluations with well-defined gold standards. Establishing such benchmarks would not only clarify the trade-offs in precision, recall, and F1-score across different approaches, but also provide researchers with evidence-based guidelines for tool selection depending on study goals. Ultimately, community-wide performance assessments could drive methodological improvements, foster the development of integrative pipelines, and enhance the reliability of functional insights derived from metagenomic data.

Newly developed but not yet widely adopted tools may offer innovative approaches, improved accuracy, or enhanced computational efficiency, addressing limitations of current methods and opening opportunities for more comprehensive functional and taxonomic analyses. For instance, DeepFRI [66] leverages deep learning strategies to perform functional annotations, achieving ~70% concordance with eggNOG, a widely recognized annotation framework. Alternatively, tools such as FlaGs [67] and FunGeCo [68] explore complementary strategies, including gene neighborhood conservation and functional genomic context, which may provide additional layers of biological insight beyond traditional annotation pipelines.

Finally, as the article is intended as a review of the current state of the art, the addition of a section dedicated to the most recent and promising trends that will determine the future of the field would be greatly beneficial. These areas include single-cell metagenomics, which bypasses assembly problems, real-time analysis of nanopore sequencer data for epidemiological surveillance and pangenome graphs for accounting for variability. A brief but comprehensive overview of these technologies would demonstrate that the review considers not only the present, but also the future.In addition, the review contains a large number of typos that need to be corrected.

Response:

We appreciate this observation regarding the future steps on metagenomics as this can serve as a prospective view of this field and as the review closure. We have included the following additional information in the revised manuscript.

Page 39 - Paragraph 6 - Lines 1639- 1682

7..- Next steps in metagenomics

Recent methodological advances are reshaping the scope of metagenomic research by addressing long standing limitations in assembly, taxonomic resolution, and the representation of microbial variability. Single-cell metagenomics has emerged as a powerful complement to metagenome MAGs, enabling the recovery of Single Amplified Genomes (SAGs) from low abundance and rare taxa. By bypassing the reliance on binning and assembly, this strategy reduces chimerism and enhances resolution at strain level, allowing accurate reconstruction of microbial populations and gene content [252,253]. Similarly, pangenome graphs are redefining reference frameworks by moving beyond linear genome representations. Instead of relying on a single reference, graph based models incorporate genetic variability such as single nucleotide variants, indels and structural rearrangements, providing a more nuanced view of intra-species diversity and enabling high resolution classification of metagenomic reads at the strain level [254]. These developments expand the analytical toolbox available for metagenomics and facilitate the exploration of microbial diversity with unprecedented precision.

The translation of these advances into clinical and epidemiological practice has become evident. Real time nanopore sequencing has demonstrated its utility in the rapid diagnosis of infections directly from patient samples. In cerebrospinal fluid, metagenomic nanopore sequencing outperformed conventional culture-based approaches, enabling pathogen identification within minutes and proving particularly useful in cases where standard diagnostic assays fail [255]. Beyond diagnosis, the capacity of long-read sequencing platforms to perform on site analysis has been highlighted as a key factor in outbreak management and epidemiological surveillance. Their ability to detect structural variants, plasmids and mobile genetic elements in near real time enhances our capacity to track antimicrobial resistance and pathogen evolution during public emergencies [256].  In parallel, single-cell metagenomics offers a route to link specific antibiotic resistance genes or metabolic functions directly to their microbial hosts, a level of resolution with clear clinical relevance for antimicrobial stewardship and personalized microbiome targeted interventions [252].

Despite these promising advances, significant challenges remain before these approaches can be broadly implemented. Single-cell metagenomics is limited by technical hurdles such as contamination, amplification bias, and uneven genome coverage, which can compromise the quality of SAGs [253]. Moreover, while single-cell workflows are becoming more scalable, they remain resource-intensive compared to bulk metagenomic sequencing. Nanopore sequencing, although invaluable for rapid diagnostics, still suffers from relatively high error rates compared to short-read technologies, which can complicate the detection of low-frequency variants or subtle strain differences [255]. Similarly, while pangenome graphs provide a conceptually robust framework, they require extensive computational resources and high-quality reference collections to build and curate graph structures. Their adoption also depends on the development of standardized methods and visualization tools that make graph-based analyses accessible to non-specialist users [254]. Taken together, these limitations emphasize that while single-cell approaches, real-time sequencing, and pangenome frameworks represent the next frontier in metagenomics, further refinement and validation are essential for their integration into routine research and clinical pipelines.

To the specific lines:

line 57: The citation "(Lema et al., 2023)" should be formatted as "[2]";

Page 2 - Paragraph 2 - Lines 54- 56

Overcoming these hurdles will be essential to fully harness the potential of computational metagenomics in translational research and clinical applications [2].

line 241: Figure 2 is not referenced in the main text, unlike the other figures;

Page 8 - Paragraph 1 - Lines 262

In the revised version Figure 2 is now referenced in section 2.2 Bioinformatic Pipelines.

These workflows are summarized in Figure 2.

lines 243-244: It is not clear where points A) and B) are shown in Figure 2;

Page 8 - Paragraph 1 - Line 262 

The points A) and B) from Figure 2 were eliminated in the revised version.

lines 246-254, 256-266, 268-289: Unnecessary italics. Please use regular font;

Page 8 - Lines 264 - 300

Italics were changed to regular font in the revised version.

lines 325-329: The text describing the MIMAG standarts is almost completely repeated on lines 337-340;

Page 12 - Paragraph 1 - Lines 370- 371

These metrics are part of the MIMAG standards mentioned above [27].

lines 329 and 340: I did not find (Bowers et al., 2017) in the list of references;

Page 11- Paragraph 2 - Line 358

Bowers et al reference is cited in the correct format in the revised version

Response: 

We appreciate the reviewer’s careful evaluation and helpful suggestions. All the points raised, including citation formatting, figure referencing and labeling, font usage, repeated text, and missing references, have been addressed in the revised manuscript. We believe these changes improve the clarity, consistency, and overall presentation of the work.

Page 13- Paragraph 5 - Line 446 - 447

Short reads oriented tools such as Bracken [45], Kraken 2 [39], Centrifuge [40] and CLARK/CLARK-S [46,47] perform k-mer based matching…

line 410: Remove the unnecessary period after "[44,45].";

Page 13- Paragraph 5 - Line 447

CLARK/CLARK-S [46,47] perform k-mer based matching against reference databases.

line 413: You probably meant "achieve", not "archive". And also "resolution", not "resolutions";

Page 14 - Paragraph 1 - Line 461

Although these tools can achieve species level resolution…

line 427: Misprint "MetaPlAn" -> "MetaPhlAn";

Page 14 - Paragraph 2 - Line 478

MetaPhlAn 4 adds support for long read sequence profiling through Minimap2-based alignment.

line 436: Unexpected line break between "DIAMOND" and "[54]";

Page 14 - Paragraph 3 - Line 483

This section of the text was removed to avoid redundancy with section 2.2.6.3 Functiona annotation.

line 451: "...aiming to bridge the gap" would be grammatically more correct than "...in aims to bridge the gap";

Page 15 - Paragraph 2 - Line 511

To address this limitation, various computational tools have been developed to leverage marker gene information aiming to bridge the gap between taxonomic profiling and functional characterization by predicting metabolic pathways and biological processes based on marker gene distributions.

line 467: Misprint "gen transfer" -> "gene transfer";

Page 15 - Paragraph 2 - Lines 527 - 528

However, this approach overlooks intra-species variability, horizontal gene transfer events and strain specific adaptations.

line 475: "which are capable of annotating" would be grammatically better than "...them being capable of annotate";

Page 15 - Paragraph 4 - Line 536 

In this regard, tools like Prokka [68] and InterProScan [69] have been tested to annotate test genomes, which are capable of annotating between 75 and 90% of the individual genomes in less than 3 minutes. 

line 483: Misprint "BlasKOALA" -> "BlastKOALA";

Page 15 - Paragraph 5 - Line 544

BlastKOALA delivers accurate annotations at the genome level, but its computational cost is substantial…

line 509: Remove the duplicated period after "previous studies";

Page 17 - Paragraph 4 - Line 626

To facilitate comparisons among such methods, Table 4 summarizes representative tools that have been widely tested in previous studies. 

line 512: The "Dataset" column lists vastly different and incompatible data sets for many tools (differing in size and origin). This makes a direct comparison of "Memory use" and "Running time" across tools impossible;

Response:

We thank the reviewer for this valuable observation. We agree that the datasets listed under the "Dataset" column differ considerably in size and origin, which indeed prevents direct comparison of memory usage and running time across tools. Our intention in including this information was not to suggest strict comparability, but rather to provide additional context regarding the type of datasets on which performance was reported. In fact, benchmarking studies that simultaneously evaluate all the tools discussed in our manuscript are lacking, and in several cases, the corresponding publications did not report standardized performance metrics. Moreover, in the specific case of functional analysis tools, robust benchmarks or reference datasets enabling systematic evaluation of computational performance and accuracy are not yet available. We have clarified this limitation in the revised version of the manuscript.

line 541: SnakeWRAP is actually implemented in Snakemake, not Nextflow;

Page 19 - Paragraph 2 - Line 656

On the other hand, tools such as SnakeWRAP [88], likewise implemented in Snakemake

line 564: "biomedicine field" would be better than "biomedicine area";

Page 19 - Paragraph 5 - Line 680

As Martins [95] has stated, metagenomic databases can be organized and classified following different criteria, to this end, and only contemplating the human biomedicine field…

line 712: "this datasets" -> "these datasets";

Page 22 - Paragraph 6 - Line 845

Metagenomic data exhibits several particularities that complicate statistical modeling. First, these datasets are inherently high dimensional…

line 715: "this features" -> "these features";

Page 22 - Paragraph 6 - Line 849

…the heterogeneous distribution of these features across hosts and the limited sequencing depth.

line 717: Misprint "in wich" -> "in which";

Page 22 - Paragraph 6 - Line 851

Moreover,  when features are detected, their counts usually exhibit a large fluctuation across samples, leading to over-dispersion, in which the variance greatly exceeds the mean.

line 732: "Variation Stabilization Transformation" -> "Variation Stabilizing Transformation";

Page 23 - Paragraph 2 - Line 867

and Variation Stabilizing Transformation (VST) transform relative abundances to a logarithmic space, eliminating the constant sum restriction. 

line 754: "data as tested Nearing (2022)" -> "data, as tested by Nearing et al. (2022)";

Page 23 - Paragraph 7 - Line 893

Nearing et al. [138] systematically…

line 916: "Receiver Operating Characteristic curve" -> "Receiver Operating Characteristic (ROC) curve";

Page 28 - Paragraph 1 - Line 1109

…. F) Receiver Operating Characteristic (ROC) curve.

line 933: "The field of machine learning is fairly vast" -> "The field of machine learning encompasses a wide range of techniques";

Page 28 - Paragraph 4 - Line 1130

The field of machine learning encompasses a wide range of techniques. It can be broadly divided into two categories:

line 986: "Table 4" -> "Table 5" ("Table 4" already exists on page 14);

Page 30 - Paragraph 3 - Line 1195

This error is fixed in the revised version

line 996: Unexpected line break between "existing databases" and "AI-based tools";

Page 30 - Paragraph 7 - Lines 1206

Limited Novelty Detection: Identifying previously uncharacterized microorganisms or genes is challenging using traditional approaches that depend on existing databases. AI-based tools address these challenges by offering enhanced capabilities [191,192].

line 998: The claim that "AI tools can process... rapidly with smaller models, significantly reducing storage and memory usage" is an overgeneralization. Although certain models are efficient, many contemporary deep learning architectures, such as large language models and transformers, are famously resource-intensive and require vast amounts of memory. This statement needs to be softened or made more precise;

Response: 

We agree that the claim was an overgeneralization and failed to capture the important distinction between the resource-intensive nature of training modern deep learning models and their efficiency in application. We have revised the "Comparison with Traditional Tools" section to provide a more precise perspective.

Page 30 - Paragraph 8 - Lines 1208 - 1214

The new text, titled “Efficiency in Application” now explicitly discusses this crucial distinction:
Efficiency in Application: The computational profile of AI-based tools presents a critical trade-off between training and application. While the process of training deep learning models is often famously resource-intensive, requiring large datasets and significant computational power, the resulting models can be highly efficient for inference. Once trained, applying a model to classify new sequences or predict a phenotype is often computationally much faster than performing traditional, per-sample alignment-based searches against large reference databases [3].

line 1346: "...the the Minimal Information" -> "...the Minimal Information";

Page 38 - Paragraph 8 - Line 1602

Along with this principles, the Genomic Standards Consortium implemented the Minimal information for Marker Genes Sequences (MIMARKS) and the Minimal Information about any Sequence

Comments on the Quality of English Language

Esteemed colleagues,
The primary issue with the manuscript is the presence of numerous typos. It is imperative that a meticulous proofreading process is undertaken to rectify the identified technical deficiencies, thereby elevating the work's overall quality and professional standards.

Response: 

We appreciate the reviewer’s insightful remark regarding the language quality of the manuscript. We carefully revised the entire text and performed a meticulous proofreading to correct typographical errors and improve its clarity, consistency, and overall professional quality.